# Three new species of *Byrsopteryx* Flint microcaddisflies from Peru (Insecta: Trichoptera) including DNA-based larval associations

Allan P.M. Santos[1] and Daniela Maeda Takiya[2]

[1] Departamento de Zoologia, Universidade Federal do Estado do Rio de Janeiro, Rio de Janeiro, Brazil
[2] Departamento de Zoologia, Universidade Federal do Rio de Janeiro, Rio de Janeiro, Brazil

## ABSTRACT

In this paper, we have described and illustrated three new species of *Byrsopteryx* from Peru: *Byrsopteryx inti*, **sp. nov.** *Byrsopteryx mamaocllo* **sp. nov.**, and *Byrsopteryx mancocapac* **sp. nov.** Larvae of the latter two were also associated to male specimens based on comparison of a fragment of COI gene and pharate male identification. *Byrsopteryx inti* **sp. nov.** and *Byrsopteryx mamaocllo* **sp. nov.** share a unique feature: a semi-dome process formed by a thickened area on male forewings. The three species can be easily identified by wing coloration and male genitalia. Furthermore, *Byrsopteryx inti* **sp. nov.** can be recognized by its sternum VIII with a median digitate process on posterior margin, slightly capitate; and by long dorsolateral processes from segment VIII, which cross each other apically in dorsal view. *Byrsopteryx mamaocllo* **sp. nov.** can be distinguished by sternum VIII bearing a pair of short, posterior, spinelike processes, which are curved inwards and bordered by a rounded, membranous structure, and by a pair of short, heavily sclerotized, dorsolateral processes. *Byrsopteryx mancocapac* **sp. nov.** can be distinguished by strong spine-like processes arising dorsally from subgenital plate and by sternum VIII with posterior margin divided into two plate-like lobes. Larvae of *B. mamaocllo* **sp. nov.** and *B. mancocapac* **sp. nov.** are similar to other *Byrsopteryx* larvae known. They can be distinguished from each other by the shape of the operculum formed by terga VIII and IX, and number of setae on the second abdominal pleurite. Maximum likelihood analyses of 20 COI sequences, including nine *Byrsopteryx* species, placed *B. inti* **sp. nov.** and *B. mamaocllo* **sp. nov.** as sister species and related to a clade including *B. gomezi*, *B. tapanti*, and *B. esparta*, while *B. mancocapac* **sp. nov.** was found as sister to *B. abrelata*. Despite the close phylogenetic relationship found between *B. inti* **sp. nov.** and *B. mamaocllo* **sp. nov.**, they are separated by 14.9% minimum K2P divergence of COI. The highest intraspecific distance observed was 1.4% for *B. mancocapac* **sp. nov.** individuals. Although the Peruvian caddisfly fauna has around 320 known species and almost a third of them are microcaddisflies, in this paper we present the first descriptions of *Byrsopteryx* species for the country.

Corresponding author
Allan P.M. Santos,
allanpms@gmail.com

## INTRODUCTION

Hydroptilidae, or microcaddisflies, constitute the most diverse family in the order Trichoptera, with over 2,500 named species in 74 genera (*Morse et al., 2019*). Microcaddisflies are recorded from all zoogeographic regions, except Antarctic, and are particularly diverse in the Neotropics, where almost a thousand species are currently known (*Holzenthal & Calor, 2017*; *Morse et al., 2019*). Despite their diversity, microcaddisflies are usually neglected by taxonomists due to their small body size (usually less than 5.0 mm), which makes difficult to manipulate specimens and to observe morphological features.

Current family classification derives mostly from the comprehensive revision provided by *Marshall (1979)*, including 46 genera known at the time. Six subfamilies are recognized in Hydroptilidae, defined as tribes by *Marshall (1979)*, all of them recorded from the Neotropics (*Holzenthal & Calor, 2017*). Leucotrichiinae are restricted to the New World, with most species recorded from the Neotropical Region. The subfamily delimitation has been revised by *Santos, Nessimian & Takiya (2016)* based on a phylogenetic hypothesis, and now it includes two tribes: Alisotrichiini and Leucotrichiini, with respectively six and ten genera. *Byrsopteryx* *Flint Jr, 1981* belongs to the Alisotrichiini and its monophyly has been consistently supported by morphological and molecular data (*Harris & Holzenthal, 1994*; *Santos, Nessimian & Takiya, 2016*).

Among Neotropical microcaddisflies, adults of *Byrsopteryx* are easily recognized by their coloration, with white spots over a black background (*Harris & Holzenthal, 1994*; *Santos & Nessimian, 2010*). While adult caddisflies are usually collected at night with light traps, adults of *Byrsopteryx* are more active during the day, and rarely flies to nocturnal traps. They also differ from most caddisfly species due to their behavior, under bright sunlight, adults of *Byrsopteryx* can be seen running fast over rocks or large leaves of riparian vegetation in a zigzag or in a more erratic way (*Flint Jr, 1981*; *Harris & Holzenthal, 1994*; *Santos & Nessimian, 2010*). They intercalate frantic running with moments when they seem frozen in a position or when they start to revolve around the axis of their own heads (*Flint Jr, 1981*; *Harris & Holzenthal, 1994*; *Santos & Nessimian, 2010*). When disturbed, they just fly off to find another place and start these movements again. Thus, adults of *Byrsopteryx* are more easily collected with aspirators or directly with a finger moistened in alcohol. They occasionally come to light and Malaise traps, but in relatively low numbers, especially compared to other Neotropical caddisflies.

*Byrsopteryx* larvae build purse-like portable cases, like those built by *Celaenotrichia* Mosely, 1934 larvae, another genus of Alisotrichiini. Larval case is weakly sealed dorsally, made mainly of silk and usually with mineral grains and/or algae filaments added (*Holzenthal & Harris, 1991*; *Santos & Nessimian, 2010*). Larvae are typically madicolous, being found in waterfalls, living in the spray and splash zone or in small streams on boulders (*Holzenthal & Harris, 1991*; *Harris & Holzenthal, 1994*; *Santos & Nessimian, 2010*). They seem to feed mainly on diatoms, which they apparently scrape from the substrate along with associated periphyton (*Holzenthal & Harris, 1991*; *Santos & Nessimian, 2010*). Pupae are fixed to the rocky substrate by a short peduncle, and are commonly found in aggregation

above the waterline in pits or small depressions (*Holzenthal & Harris, 1991*; *Santos & Nessimian, 2010*).

Currently, 16 species are assigned to *Byrsopteryx* and they occur from southern Mexico; in Central America, including the Caribbean; through northwestern and southeastern South America (*Holzenthal & Calor, 2017*). Larvae of four species are currently known: *B. abrelata Harris & Holzenthal, 1994*, *B. carioca Santos & Nessimian, 2010*, and *B. espinhosa Harris & Holzenthal, 1994* from Brazil (*Santos & Nessimian, 2010*), and *B. mirifica Flint Jr, 1981* from Venezuela (*Flint Jr, 1981*; *Holzenthal & Harris, 1991*). Just recently, typical larvae of *Byrsopteryx* were recorded from Colombia (*Vásquez-Ramos, Osorio-Ramírez & Caro-Caro, 2020*). *Santos, Nessimian & Takiya (2016)* listed three undetermined *Byrsopteryx* species from Peru among the examined material used for phylogenetic analyses, but no species of *Byrsopteryx* is recorded from the country so far. Peru has a highly diverse fauna of caddisflies, with around 320 known species, almost a third of them being microcaddisflies (*Holzenthal & Calor, 2017*). Important works on this fauna include those of *Flint Jr (1975)*, *Flint Jr (1980)*, *Flint Jr (1996)* and *Flint Jr & Reyes (1991)*, and those focusing on hydroptilids of *Flint Jr & Bueno-Soria (1998)*, *Flint Jr & Bueno-Soria (1999)*, *Harris & Davenport (1992)* and *Harris & Davenport (1999)*. However, many new species of caddisflies from Peru remain in collections to be described, indicating that this fauna is still very underworked. Here we describe three new species of *Byrsopteryx* from Peru, including larval descriptions for two of them. Larval associations for these species are based on comparison of a fragment of the mitochondrial gene cytochrome oxidase I (COI) from males and larvae and also, for one of them, comparison among cases and structures of pharate males and larvae.

## MATERIALS & METHODS

### Specimen collecting and morphological study

Adults and immatures of *Byrsopteryx* studied here were collected with aspirators, with finger moistened in alcohol (adults) or with aid of entomological forceps (immatures). In addition, adults were also collected with a Malaise trap set at one of the collecting sites. Specimens were collected under permit RD-0297-2012-AG-DGFFS-DGEFFS, issued by Dirección General Forestal y de Fauna Silvestre, Peru. Larvae and most adults were fixed and preserved in 96% ethanol, however some adults were killed in jars with ethyl acetate and pinned dry, to better preserve coloration. *Byrsopteryx* specimens were found in four sites, in Cusco and Puno provinces, Peru. Collecting sites exhibited the typical environment for *Byrsopteryx* species: rocky streams and small waterfalls (Fig. 1).

Male and female genital structures were analyzed after the clearing procedure with a heated solution of 10% KOH, as described in *Ross (1944)*. Abdomens were mounted in temporary slides and observed under a compound microscope equipped with camera lucida. Then, pencil sketches were made and used as templates in Adobe Illustrator (Adobe Systems Inc.) to create vector graphics, as detailed in *Holzenthal (2008)*. Adult male holotypes and larvae were photographed with a digital camera attached to a Leica stereomicroscope. Photographs at different focal planes were obtained and then stacked with Leica Application Suite or Helicon focus Lite software and then edited in Adobe

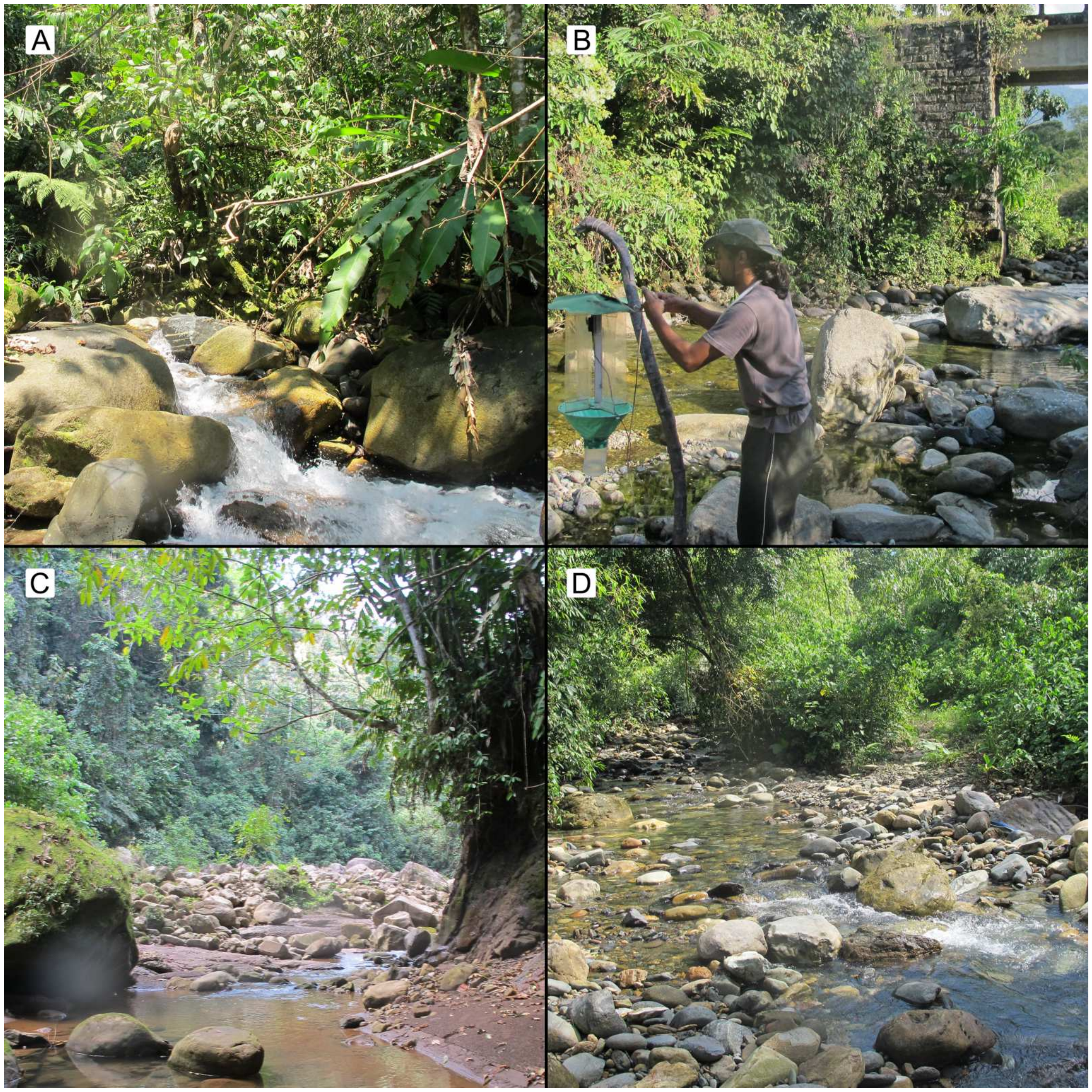

**Figure 1 Collecting localities of specimens of new *Byrsopteryx* species described here.** (A) Tributary of Río Araza, Cusco, Peru. (B) Pennsylvania trap being set up by APMS near Puente Saucipata, close to a tributary of Río Araza, Cusco, Peru. (C) Tributary of Río Araza, near Puente La Cigarra, Cusco, Peru. (D) Stream near Quincemil, Cusco, Peru.

Photoshop (Adobe Systems Inc.). After observation, abdomens were stored in microvials with ethanol, for specimens in alcohol, or glycerin for pinned specimens. Terminology used here follows that of *Harris & Holzenthal (1994)*, except for the inferior appendages, for which we used the interpretation of *Santos, Nessimian & Takiya (2016)*. Holotypes of the newly described species are deposited at Museo de Historia Natural, Universidad Nacional Mayor de San Marcos, Lima (MUSM). Paratypes are in the same institution and also in Coleção Prof. José Alfredo Pinheiro Dutra, Departamento de Zoologia, Universidade Federal do Rio de Janeiro, Rio de Janeiro (DZRJ) and Coleção de Invertebrados, Instituto Nacional de Pesquisas da Amazônia, Manaus (INPA). Extracted DNA solutions are deposited at DZRJ.

## DNA extraction, amplification, and sequencing

DNA was extracted from the entire body of specimens using the Wizard SV Genomic DNA Purification System (Promega Corporation, Madison, WI, USA), without tissue maceration. After extraction, specimens were returned to ethanol and deposited in the original collection as a DNA voucher. COI fragments were amplified using different pair of primers (*Folmer et al., 1994*; *Simon et al., 1994*): HCO-2198 (5′-TAAACTTCAGGGTGACCAAAAAATCA) in combination with LCO- 1490 (5′-GGTCAACAAATCATAAAGATATTGG); HCO-2198 with C1-J-1718 (5′-GGAGGATTTGGAAATTGATTAGTTCC); or Ron (5′-GGATCACCTGATA-TAGCATTCCC) and Nancy (5′-CCCGGTAAAATTAAAATATAAACTTC).

Polymerase chain reaction (PCR) conditions were as follows: when using HCO-2198 and LCO-1490, initial denaturation at 95 °C for 3 min; 5 cycles of denaturation at 95 °C for 1 min, annealing at 45 °C for 1,5 min, and extension at 72 °C for 1 min, then 35 cycles of 95 °C for 40 s, 51 °C for 1 min, and 72 °C for 1 min; and final extension at 72 °C for 7 min. When using other primer combinations, initial denaturation at 94 °C for 3 min; then 35 cycles of 94 °C for 1 min, 50 °C for 1 min, and 72 °C for 2 min; and final extension at 72 °C for 7 min. PCR products were sent to Macrogen Inc., Seoul, for purification and sequencing reactions.

## Phylogenetic analysis

Forward and reverse electropherograms of each sample were assembled and manually edited in Sequencher 4.1 (Gene Codes, Ann Arbor, Michigan, USA). Consensus sequences were verified with the Blast tool in GenBank to check for contamination. Additional sequences of *Byrsopteryx* and of *Celaenotrichia edwardsi* Mosely, 1934 as outgroup, were obtained from Bold Systems (Table 1). COI sequences were aligned using ClustalW algorithm implemented in MEGA X (*Kumar et al., 2018*) and translated into amino-acid sequences to check for stop codons. Final alignment resulted in a matrix with 653 bp and 20 sequences (File S1). All COI sequences are available in GenBank (accession numbers are provided in Table 1). MEGA X was also used to calculate Kimura 2-Parameter (K2P) divergences (*Kimura, 1980*), with pairwise deletion of missing information. A maximum likelihood analysis was conducted in RAxML 8.2.11 (*Stamatakis, 2014*) with 500 search replicates and model GTR+G+I, as selected in jModeltest 2 (*Darriba et al., 2012*) using Akaike Information Criterion (*Akaike, 1974*). Branch support tree was assessed with 1,000 pseudoreplicates of non-parametric bootstrap (BS, *Felsenstein, 1985*).

## New species names

The electronic version of this article in Portable Document Format (PDF) will represent a published work according to the International Commission on Zoological Nomenclature (ICZN), and hence the new names contained in the electronic version are effectively published under that Code from the electronic edition alone. This published work and the nomenclatural acts it contains have been registered in ZooBank, the online registration system for the ICZN. The ZooBank LSIDs (Life Science Identifiers) can be resolved and the associated information viewed through any standard web browser

**Table 1 Voucher specimen and GenBank accession numbers.** Species of *Byrsopteryx* and other Leucotrichiinae with COI sequences used in this study, with respective information of voucher specimen and GenBank accession numbers.

| Species | Voucher code | Life stage/ adult gender | Country and state/province | GenBank accession number |
|---|---|---|---|---|
| *B. espinhosa* Harris & Holzenthal, 1994 | ENT0005 | | Brazil, Rio de Janeiro | KU094932 |
| *B. carioca* Santos & Nessimian, 2010 | ENT0056 | | Brazil, Rio de Janeiro | KU094939 |
| *B. abrelata* Harris & Holzenthal, 1994 | ENT0068 | | Brazil, Rio de Janeiro | KU094942 |
| *B. esparta* Harris & Holzenthal, 1994 | ENT0123 | | Costa Rica, Puntarenas | KU094953 |
| *B. inti* sp. nov. | ENT0702 | male | Peru, Cusco | KU094974 |
| *B. mamaocllo* sp. nov. | ENT0703 | male | Peru, Cusco | KU094975 |
| *B. mamaocllo* sp. nov. | ENT5516 | male | Peru, Cusco | OK340604 |
| *B. mamaocllo* sp. nov. | ENT5519 | larva | Peru, Cusco | OK340605 |
| *B. mancocapac* sp. nov. | ENT0704 | male | Peru, Cusco | KU094976 |
| *B. mancocapac* sp. nov. | ENT5408 | larva | Peru, Cusco | OK340606 |
| *B. mancocapac* sp. nov. | ENT5409 | larva | Peru, Cusco | OK340607 |
| *B. mancocapac* sp. nov. | ENT5410 | larva | Peru, Cusco | OK340608 |
| *B. mancocapac* sp. nov. | ENT5411 | larva | Peru, Cusco | OK340609 |
| *B. mancocapac* sp. nov. | ENT5413 | larva | Peru, Cusco | OK340610 |
| *B. mancocapac* sp. nov. | ENT5414 | larva | Peru, Cusco | OK340611 |
| *B. mancocapac* sp. nov. | ENT5494 | male | Peru, Cusco | OK340612 |
| *B. gomezi* Harris & Holzenthal, 1994 | UMSP000035603 | | Costa Rica, Puntarenas | KX107513 |
| *B. gomezi* Harris & Holzenthal, 1994 | – | | – | AF436490 |
| *B. tapanti* Harris & Holzenthal, 1994 | UMSP000075777 | | Costa Rica, Cartago | HQ971757 |
| *Celaenotrichia edwardsi* Mosely, 1934 | UMSP000038367 | | Chile, Araucania | HQ971758 |

by appending the LSID to the prefix http://zoobank.org/. The LSID for this publication is: urn:lsid:zoobank.org:pub:71531CB9-F919-4DB9-8885-B8207F0A82F4. The online version of this work is archived and available from the following digital repositories: PeerJ, PubMed Central and CLOCKSS.

# RESULTS

## Phylogenetic analysis and species divergences

The maximum likelihood tree (-lnL= 3322.784548, Fig. 2) recovered two of the new species described herein, *B. inti* **sp. nov.** and *B. mamaocllo* **sp. nov.**, as sister species (BS = 79) and related to a clade (BS = 52) including *B. gomezi*, *B. tapanti*, and *B. esparta*. The other new species described, *B. mancocapac* **sp. nov.**, was found as sister to *B. abrelata*, but with no significant bootstrap support.

Concerning species interspecific COI K2P divergences, *Byrsopteryx* species varied from 12.3–26.1% (see Table S1), minimum divergence (K2P distances) was between *B. tapanti* and *B. gomezi*, both from Costa Rica. Among the Peruvian species, minimum interspecific divergence was 14.9%, between *B. inti* **sp. nov.** and *B. mamaocllo* **sp. nov.**. Furthermore, maximum K2P intraspecific distances observed were 1.4% for *B. mancocapac* **sp. nov.** ($n = 8$), 0.7% for *B. mamaocllo* **sp. nov.** ($n = 3$), and 0% for *B. gomezi* ($n = 2$). Based on

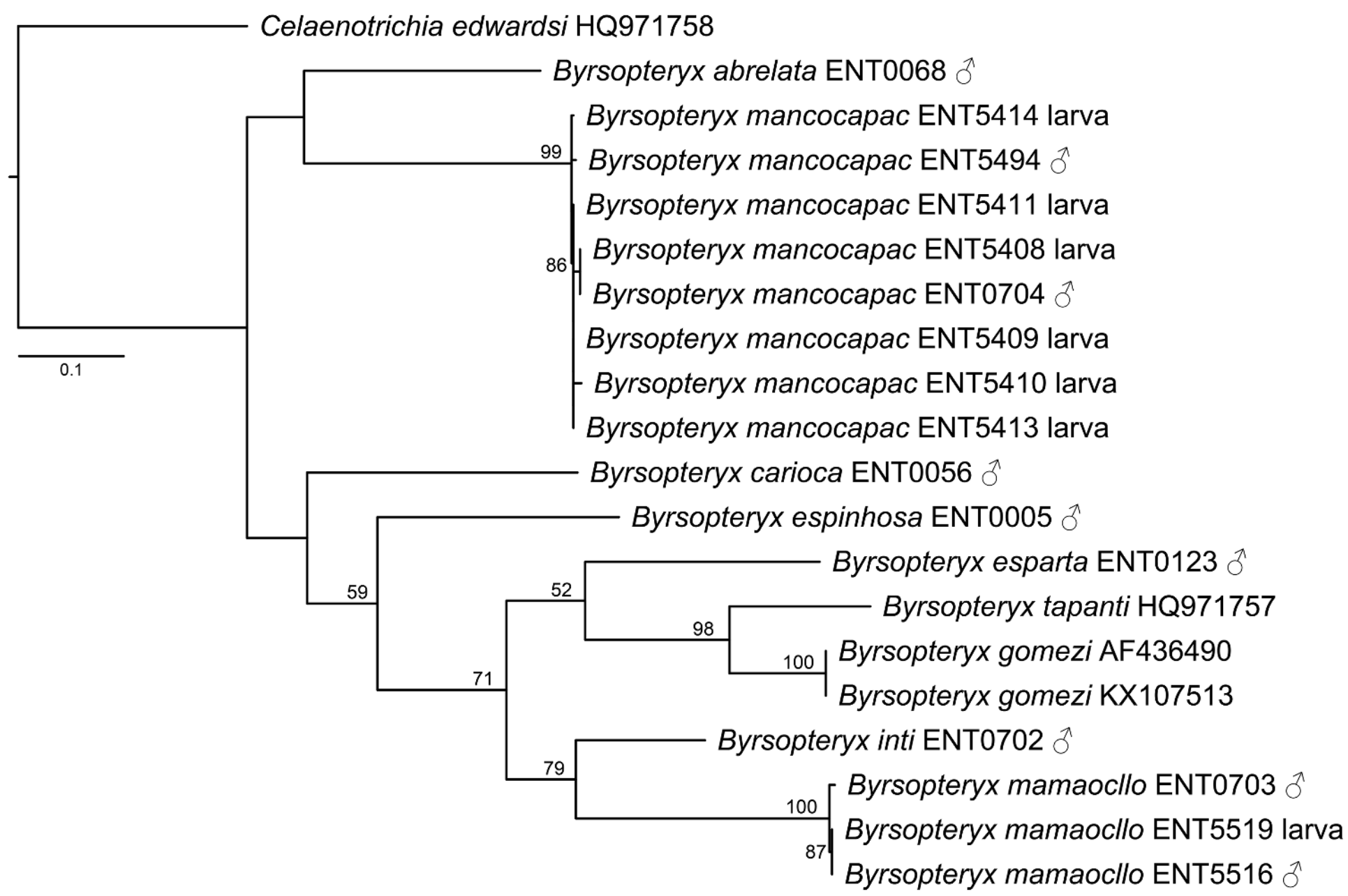

**Figure 2** **Maximum likelihood tree of COI sequences of *Byrsopteryx*.** Maximum likelihood (-lnL = 3322.784548) tree of COI sequences of *Byrsopteryx* analyzed under GTR+G+I. Numbers above branches are bootstrap percentages. Details of specimen vouchers are in Table 1.

these divergences and phylogenetic placement (Fig. 2), we were able to associate males and larval specimens of *B. mamaocllo* **sp. nov.** and *B. mancocapac* **sp. nov.**

## Taxonomy

### *Byrsopteryx inti* new species

urn:lsid:zoobank.org:act:9B5FAC2D-A98F-4B88-97BC-CE7CA5105535

- *Byrsopteryx* sp. PE1 *Santos, Nessimian & Takiya, 2016*:461. Phylogenetic placement.

Figs. 3–5

**Description.** Adult male. *Coloration*. General color dark brown; head dorsum and antennal base densely covered with white setae; mesoscutum mostly covered with white setae; forewing with four distinct maculae of white setae: an oval band at base of medial area, a trapezoidal macula near midcostal margin, a small subapical spot near posterior margin of wing, and an apical spot (Fig. 3). *Length*. Total length 2.4–2.6 mm ($n = 3$). *Head*. Unmodified. Antennae 19-articulated. Ocelli 3. Maxillary palpi 5-articulated; articles I and II very short and globular, article III very long, as long as articles IV and V combined; articles IV and V with similar lengths, each one about twice as long as wide. *Thorax*. Forewing venation strongly reduced; with small, oblique basal lobe; line of weakness distinct, bordered by row of very long, thin setae; with semi-dome formed by a thickening of wing membrane, to which apices of very long setae converge; retinaculum distinct (Fig. 4A, Fig. 4B). Hind wing venation reduced; frenulum distinct, with row of four short, hooked setae (Fig. 4C). Tibial spur formula 0, 3, 4. *Abdomen*. Segment VII without ventromesal process. *Male genitalia*. Segment VIII shorter dorsally than ventrally. Sternum with median

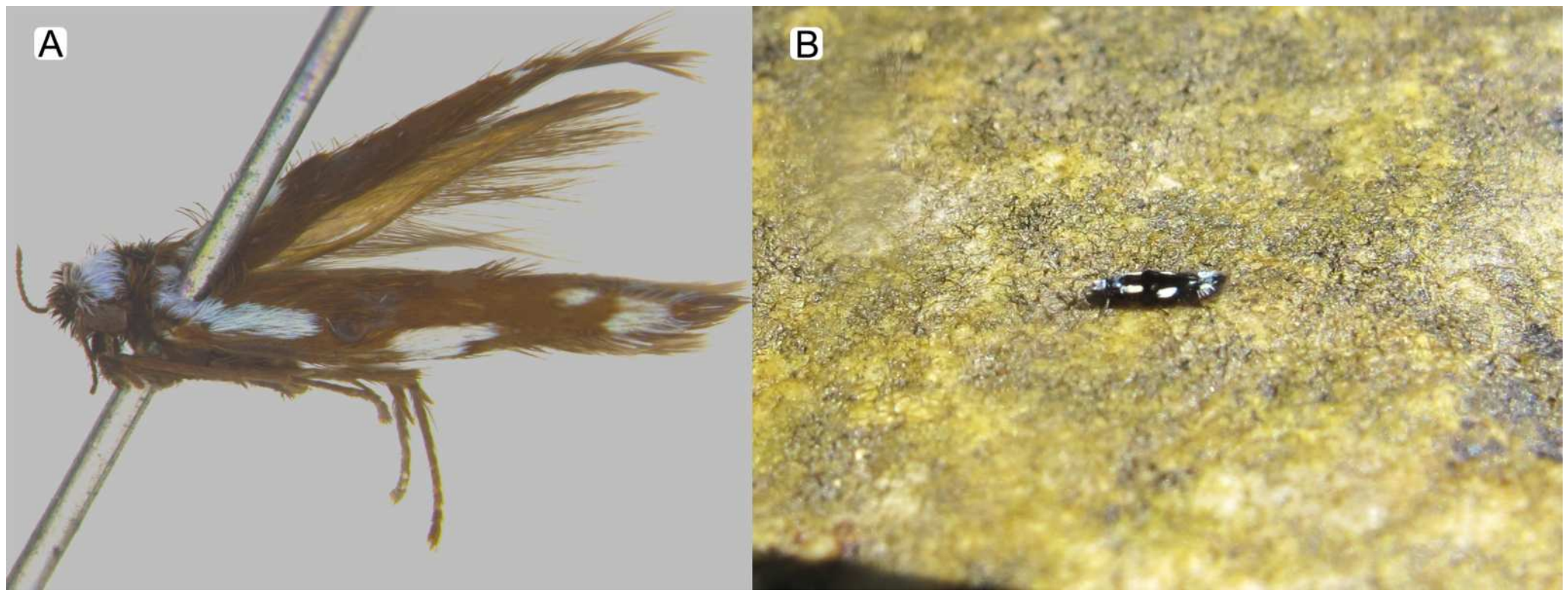

**Figure 3** ***Byrsopteryx inti* sp. nov., adult.** (A) Holotype male (pinned), lateral habitus. (B) Live adult on a rocky surface.

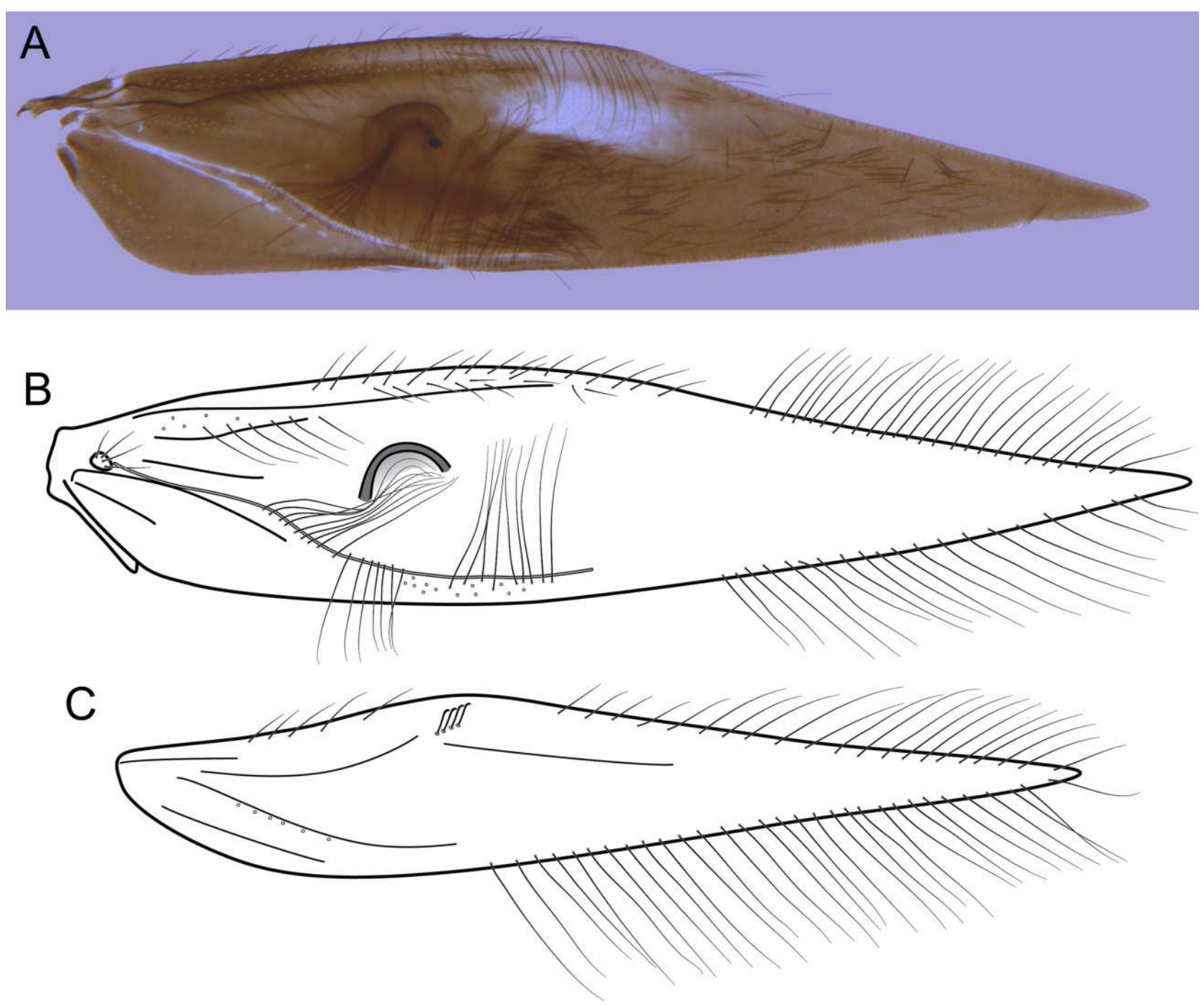

**Figure 4** ***Byrsopteryx inti* sp. nov., male wings (paratype).** (A) Forewing, showing semi-dome process. (B) Forewing. (C) Hind wing.

digitate process posteriorly, slightly capitate (Fig. 5A); with pair of posteroventral processes, diverging apically in ventral view (Fig. 5A), slightly upturned in lateral view, each one with a thin rounded membrane at apex (Figs. 5B, 5D); and with pair of dorsolateral processes, crossing each other apically in dorsal view (Fig. 5C), upturned in lateral view (Fig. 5D). Very long stout setae covering apical portions of both sternum and tergum. Segment IX recessed within segment VIII, projecting anteriorly through posterior portion of segment VII; open

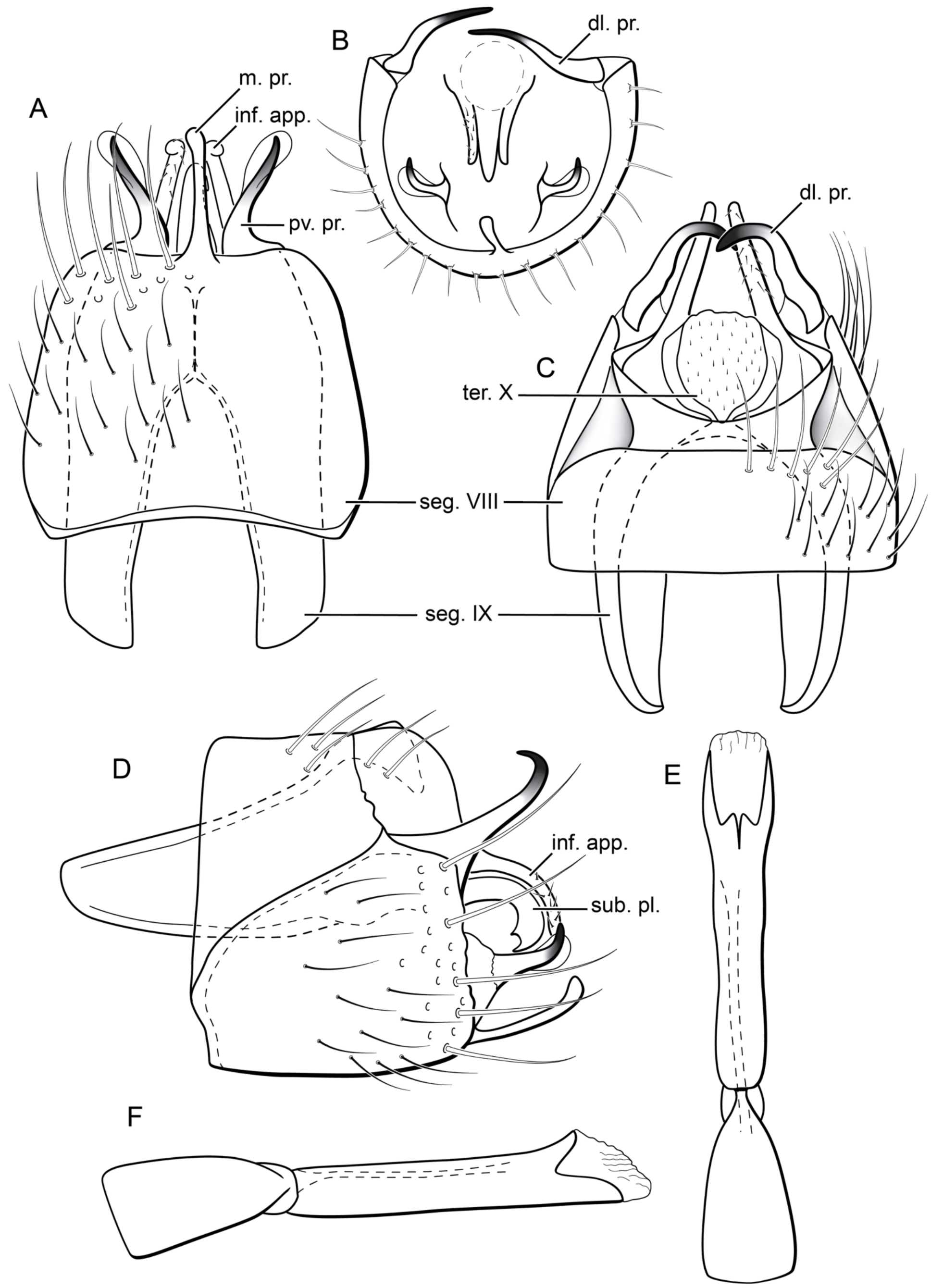

**Figure 5** ***Byrsopteryx inti* sp. nov., male genitalia (holotype).** (A) Ventral view. (B) Sternum VIII, subgenital plate, and inferior appendages, caudal view. (C) Dorsal view. (D) Lateral view. (E) Phallus, dorsal view. (F) Phallus, lateral view. Abbreviations: seg. segment; m. pr. median process (segment VIII); pv. pr., posteroventral process (segment VIII); dl. pr., dorsolateral process (segment VIII); ter. X, tergum X; sub. pl., subgenital plate; inf. app., inferior appendage.

ventrally; with deep mesal incision anteriorly in dorsal and ventral views (Figs. 5A, 5C); narrowing anteriorly in lateral view (Fig. 5D). Subgenital plate slender, with rounded apex in ventral view (Fig. 5A), downturned and with apical semicircular incision (Fig. 5D). Inferior appendages positioned dorsolaterally; elongate, digitate, converging apically in ventral and dorsal view (Figs. 5A, 5C); downturned in lateral view (Fig. 5D); covered with very short setae. Tergum X membranous; covered with minute setae; with sclerotized lateral bars. Phallus tubular, with slight constriction between basal and apical portions; basal portion short, about half length of apical portion and slightly wider; sclerotized region of apex with deep median incision in dorsal view, with pair of short triangular projections at basal margin of the incision (Fig. 5E), with a V-shaped incision in lateral view (Fig. 5F); membranous at apex.

Adult female. Unknown.

Larva. Unknown.

**Type material.** Holotype male. PERU: Cusco, 19 rd km W Quincemil, Río Araza tributary. 13°20′10″S 70°50′57″W 874 m. 23-28.viii.2012. Malaise trap. RR Cavichioli, JA Rafael, APM Santos, DM Takiya leg. Alcohol (MUSM). Paratypes. Same data as holotype, except 1 male, pinned (DZRJ); same data as holotype, manual collecting, 2 males, in alcohol (MUSM), 1 male, in alcohol (DZRJ).

**Etymology.** The species name (used as a noun in apposition) is an allusion to the popular Inca god Apu Inti, the sun god. The Inca people appeared in the Andes region and established the Inca Empire in pre-Colombian America. Cusco, where the type material comes from, was the center of the Inca Empire.

**Distribution.** Peru (Cusco).

**Remarks.** Among the three new species described here from Peru, *Byrsopteryx inti* **sp. nov.** was the most rarely collected. Only five male specimens were collected and, although *Byrsopteryx* larvae and adults were abundantly seen and collected in explored localities, both female and immature specimens of this new species were not found and remain unknown.

Males of *B. inti* **sp. nov.** share with those of *B. mamaocllo* **sp. nov.** a feature not observed in any other species in the genus: presence of a sclerotized semi-dome on forewings (Figs. 3A, 4A). Also, the general aspect of the male genitalia of these two new species is also most similar (Figs. 5B, 5D), in that they share both a pair of dorsolateral and a pair of posteroventral processes on sternum VIII. Other *Byrsopteryx* species that also have dorsolateral processes on segment VIII are *B. chaconi* *Harris & Holzenthal, 1994*, *B. cuchilla* *Harris & Holzenthal, 1994*, *B. esparta* *Harris & Holzenthal, 1994*, *B. solisi* *Harris & Holzenthal, 1994*, *B. tapanti* *Harris & Holzenthal, 1994*, and *B. tica* *Harris & Holzenthal, 1994*. However, *B. inti* **sp. nov.** can be further distinguished from all others based on (1) the sternum VIII with a median digitate process, slightly capitate posteriorly (Fig. 5A); and (2) longer dorsolateral processes from segment VIII, which cross each other apically in dorsal view (Fig. 5C).

### *Byrsopteryx mamaocllo* new species

urn:lsid:zoobank.org:act:2F8A72D9-FE9C-4C82-BFA4-26238FBEEBB7

- *Byrsopteryx* sp. PE2 *Santos, Nessimian & Takiya, 2016*:461. Phylogenetic placement.
Figs. 6–10

**Description.** Adult male. *Coloration*. General color dark brown; head dorsum and antennal base densely covered with white setae; mesoscutum mostly covered with white setae; forewing with three maculae of white setae, a longitudinal band along basal third of costal margin, a subapical spot near posterior margin of wing, and an apical spot (Fig. 6). *Length*. Total length 2.6–3.0 mm ($n = 15$). *Head*. Unmodified. Antennae 20-articulated. Ocelli 3. Maxillary palpi 5-articulated; articles I and II very short and globular, article III very long, as long as articles IV and V combined; articles IV and V with similar lengths, each one about twice as long as wide. *Thorax*. Forewing venation strongly reduced; with small, oblique basal lobe; line of weakness distinct, bordered by row of very long, thin setae; with semi-dome formed by thickening of wing membrane (absent in females), to which apices of very long setae converge; retinaculum distinct (Fig. 7A, Fig. 7B). Hind wing venation reduced; frenulum distinct, with row of five short, hooked setae (Fig. 7C). Tibial spur formula 0, 3, 4. *Abdomen*. Segment VII without ventromesal process. *Male genitalia*. Segment VIII shorter dorsally than ventrally. Sternum with a pair of short spinelike posteroventral processes, curved inwards in ventral and caudal views (Figs. 8A, 8B), upturned in lateral view (Fig. 8D), each one with a thin rounded membrane; and with pair of short dorsolateral digitate processes, slightly upturned in lateral view (Fig. 8D). Very long stout setae covering apical portions of both sternum and tergum. Segment IX recessed within segment VIII, projecting anteriorly through posterior portion of segment VII; with deep mesal incision anteriorly in ventral and dorsal views (Figs. 8A, 8C); narrowing anteriorly in lateral view (Fig. 8D). Subgenital plate subtriangular in ventral and dorsal views, apex with a small darkened digitate projection (Figs. 8A, 8C); apex downturned in lateral view (Fig. 8D). Inferior appendages positioned dorsolaterally; elongate, almost straight, parallel in dorsal and ventral views (Figs. 8A, 8C); apex downturned in lateral view (Fig. 8D); covered with very short setae. Tergum X membranous; covered with minute setae or sensilla; with sclerotized lateral bars. Phallus tubular; with slight constriction between basal and apical portions; basal portion short, about half length of apical portion; sclerotized region of apex rounded in dorsal view, with a short bifid projection subapically in dorsal view (Fig. 8E), trifid in lateral view (Fig. 8F); membranous at apex.

Adult female. Coloration and general morphology of head and thorax as in male, except forewing simple, without semi-dome structure or any other modification. *Length*. Total length 2.8–3.8 mm ($n = 13$). *Abdomen*. Segment VI without ventromesal process. Segment VII elongate, ventrally with short plumose setae (Fig. 9A). Segment VIII approximately as long as wide, with row of long setae on posterior margin, internally with pair of elongate lateral apodemes extending to middle of segment VII (Fig. 9A). Segment IX short, slightly sclerotized, internally with pair of elongate lateral apodemes extending to anterior margin of segment VIII (Fig. 9A). Segment X very short, narrowed apically, with a pair of digitate papillae (Fig. 9A). Vaginal apparatus slightly sclerotized, inconspicuous in cleared specimens; elongate, with anterior region rounded (Fig. 9B).

Larva (final instar). Length 1.8–2.5 mm ($n = 29$). Head brown to dark brown, unpigmented around eyes; quadrangular; frontoclypeal and coronal sutures indistinct;

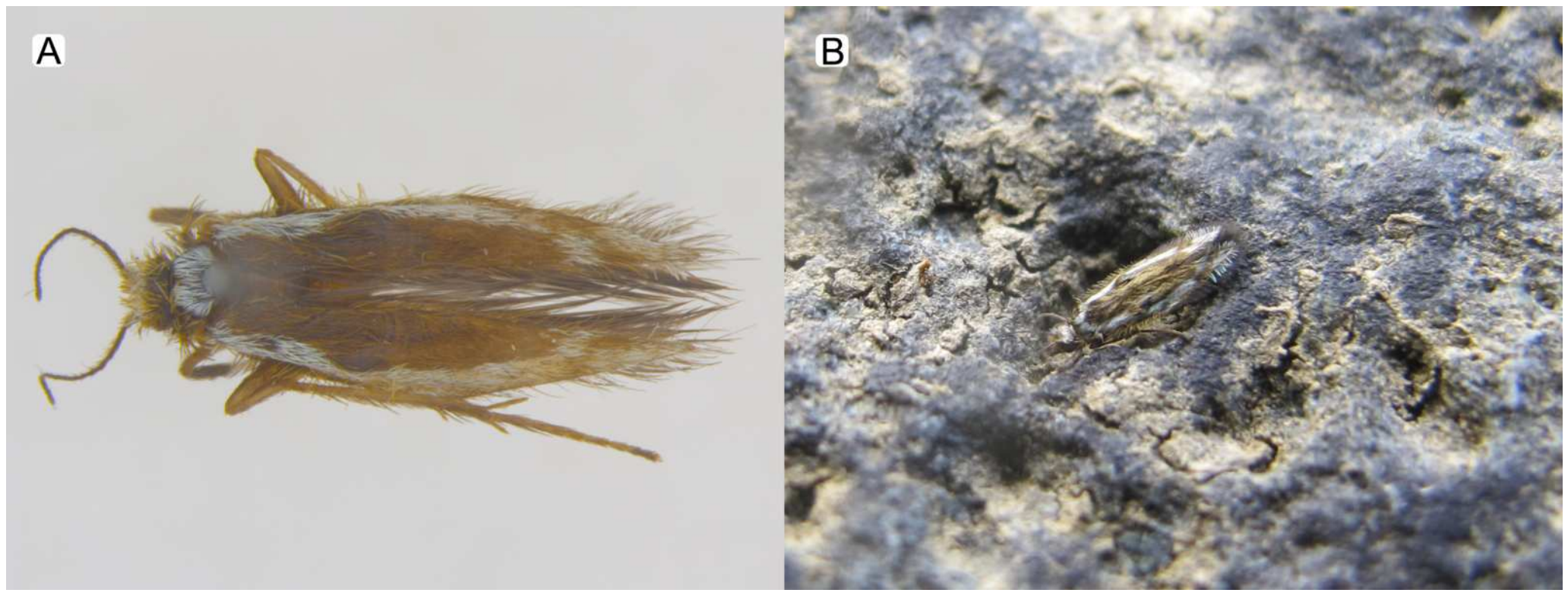

**Figure 6** ***Byrsopteryx mamaocllo* sp. nov., adult.** (A) Holotype male (pinned), dorsal habitus. (B) Live adult in a small pit over rocky surface.

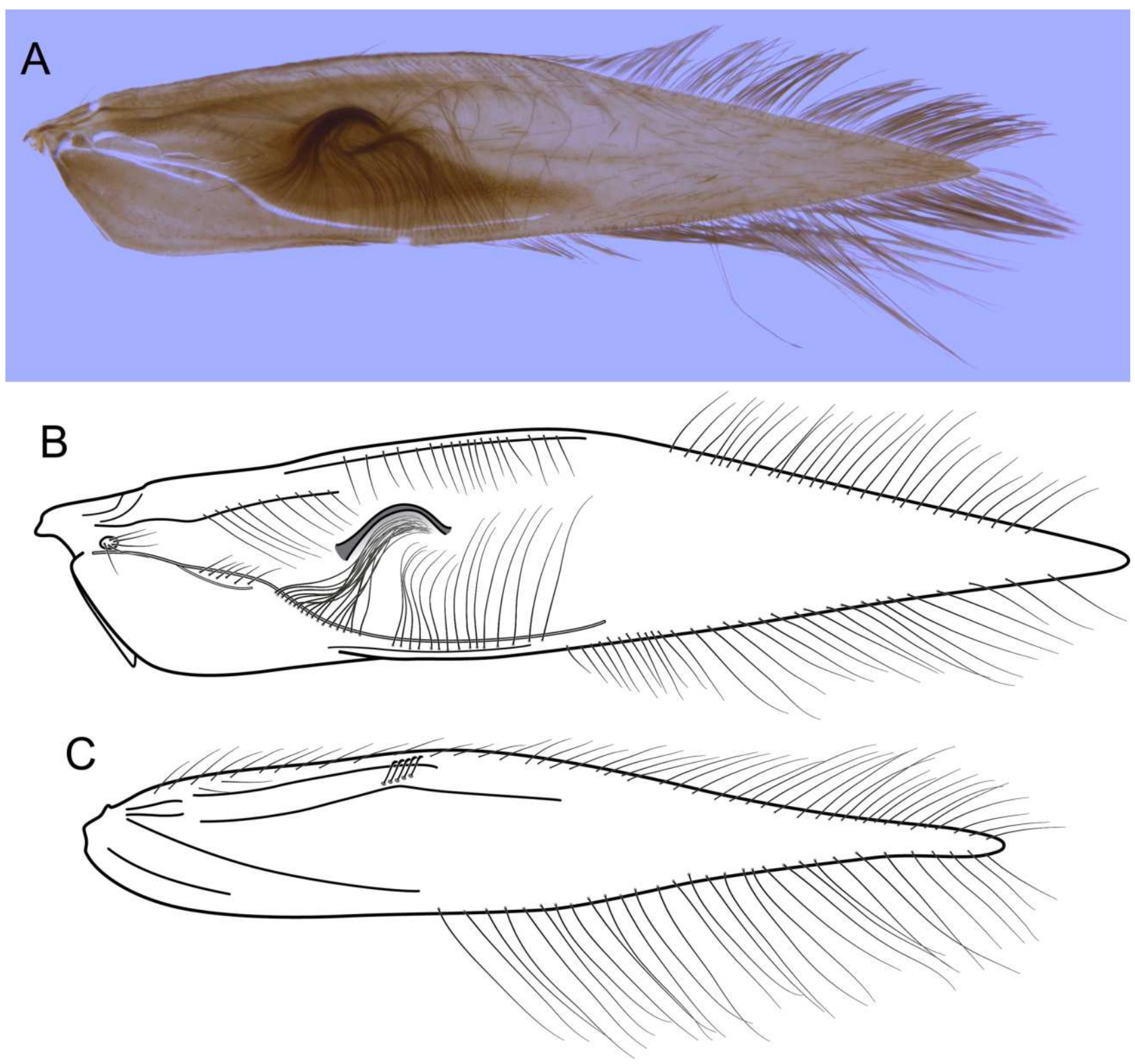

**Figure 7** ***Byrsopteryx mamaocllo* sp. nov., male wings (paratype).** (A) Forewing, showing semi-dome process. (B) Forewing. (C) Hind wing.

dorsum with reticulate pattern formed by microscopic setae; setae 9 very long; antennae 1-articulated, apical seta indistinct; mandibles without distinct teeth. Thoracic nota strongly sclerotized, brown, with lateral and posterior margins dark brown, with short stout setae (Fig. 10); pro- and mesonota with transverse mesal sclerotized ridge, less developed on

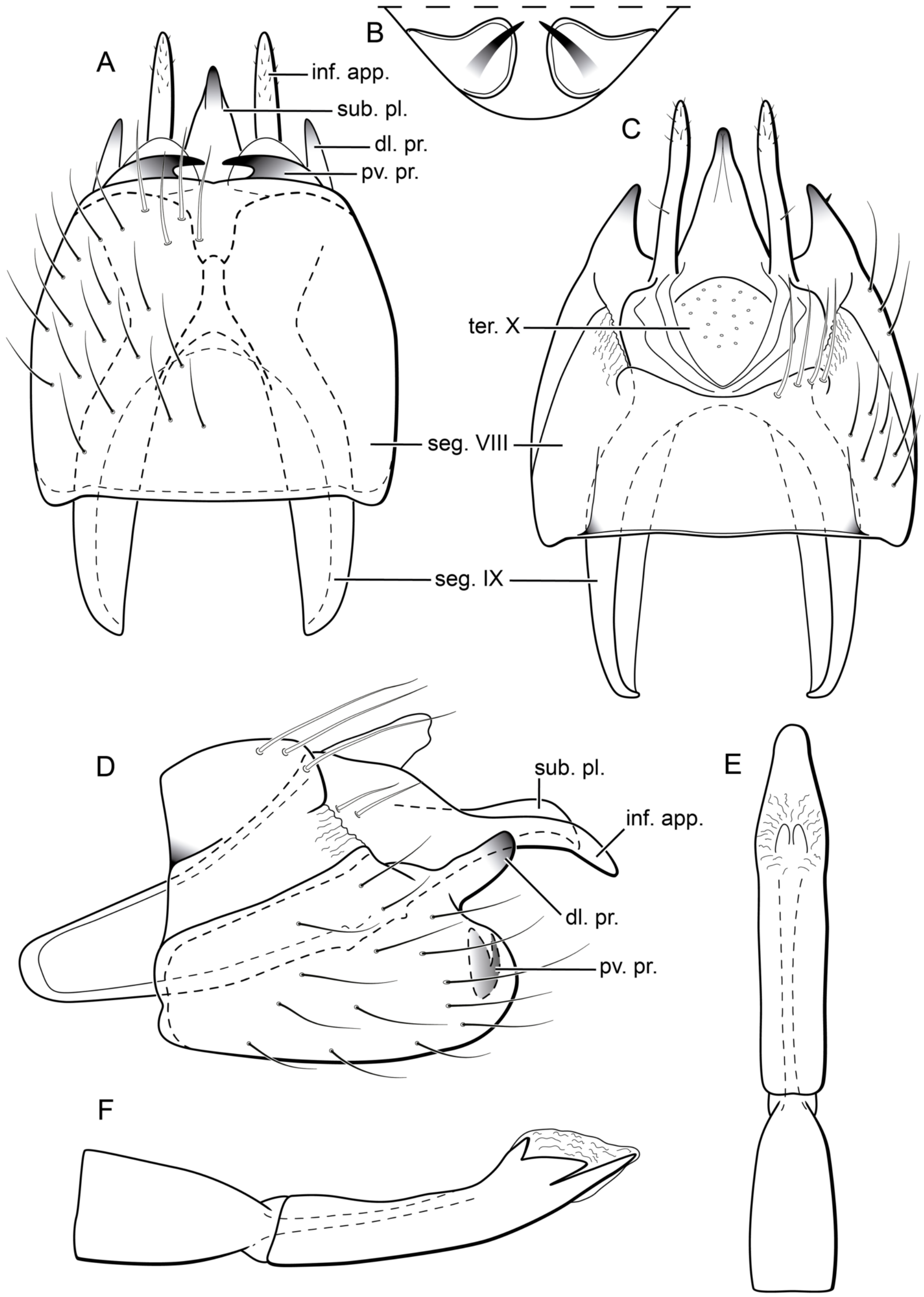

**Figure 8** ***Byrsopteryx mamaocllo*** **sp. nov., male genitalia (holotype).** (A) Ventral view. (B) Posteroventral processes of segment VIII, caudal view. (C) Dorsal view. (D) lateral view. (E) Phallus, dorsal view. (F) Phallus, lateral view. Abbreviations: seg., segment; pv. pr., posteroventral process (segment VIII); dl. pr., dorsolateral process (segment VIII); ter. X, tergum X; sub. pl., subgenital plate; inf. app., inferior appendage.

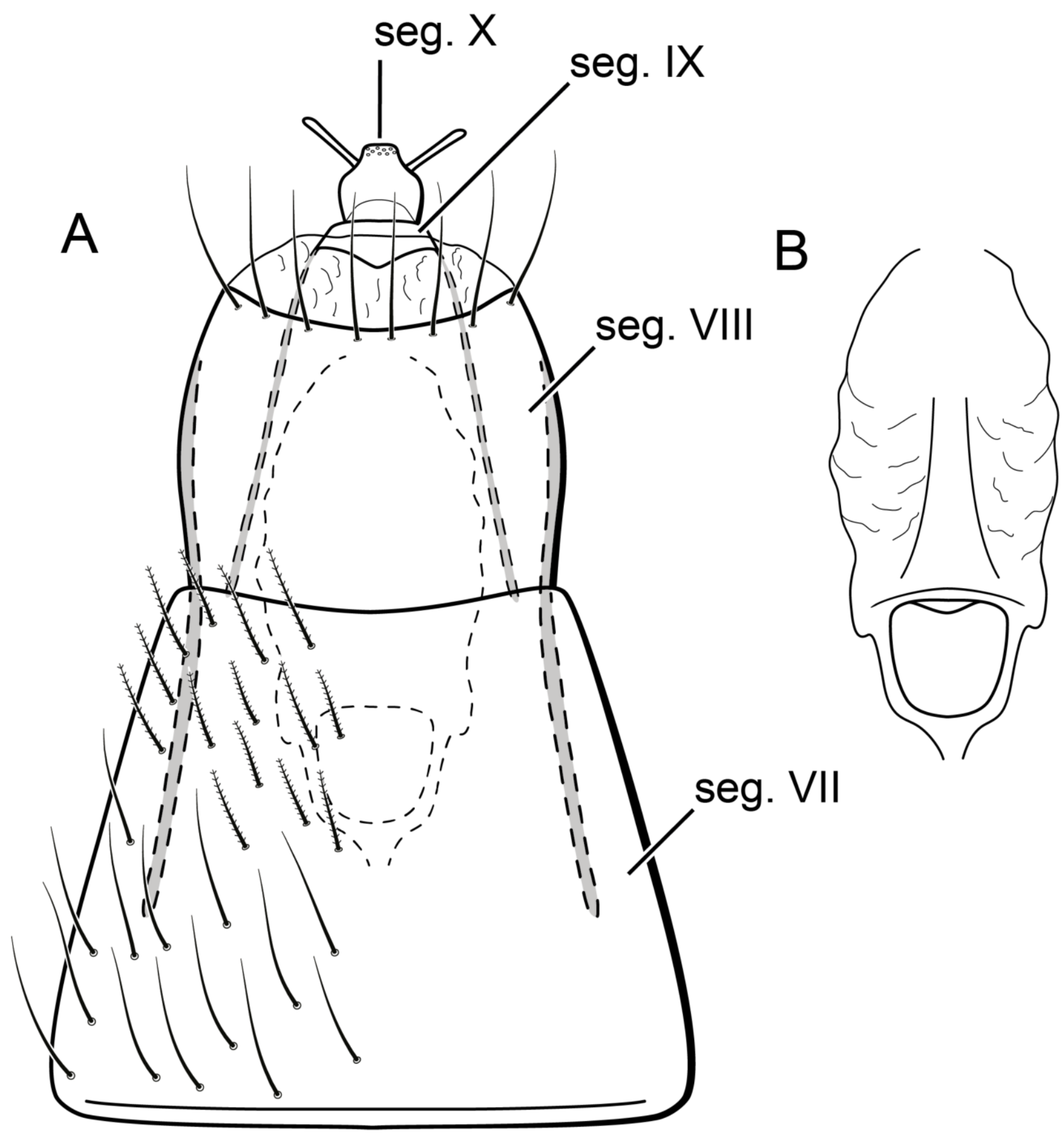

**Figure 9** ***Byrsopteryx mamaocllo* sp. nov., female genitalia.** (A) Segments VII, VIII, IX, and X, ventral view. (B) Vaginal apparatus, ventral view. Abbreviation: seg., segment.

metanotum; pronotum with middorsal ecdysial line, pair of anterolateral depressed areas (Fig. 10C); meso- and metanota slightly broader than long, without middorsal ecdysial lines (Fig. 10C); meso- and metathoracic pleural sclerite large, sclerotized; ventrally, prothorax with two pairs of thin, elongate sclerites, which converge medially; with a ventral thin intersegmental sclerite between meso- and metathorax; thoracic legs short, stout, similar in size, shape, and setation (Fig. 10B). Abdomen slightly compressed (Figs. 10B, 10C); all abdominal segments with sclerotized tergites: segments I and II with large tergite, covering most of dorsal area of the segment (Fig. 10C); segments III–VII with transverse tergite, lightly sclerotized, bearing many setae; segments VIII and IX with platelike tergite, heavily sclerotized, setose, forming a dorsal, almost circular operculum (Fig. 10D), which has a typical color pattern, with tergites dark brown with a pair of oval lighter areas (Fig. 10D); operculum closing posterior opening of the case. Pleurites of segments I and II small, with three and two setae respectively (Fig. 10B, in detail); those of segments III–VIII absent, but region with two small setae each one. Anal proleg short, setose with pair of long posterior setae; anal claw stout, curved to approximately 90° , with basal peglike setae.

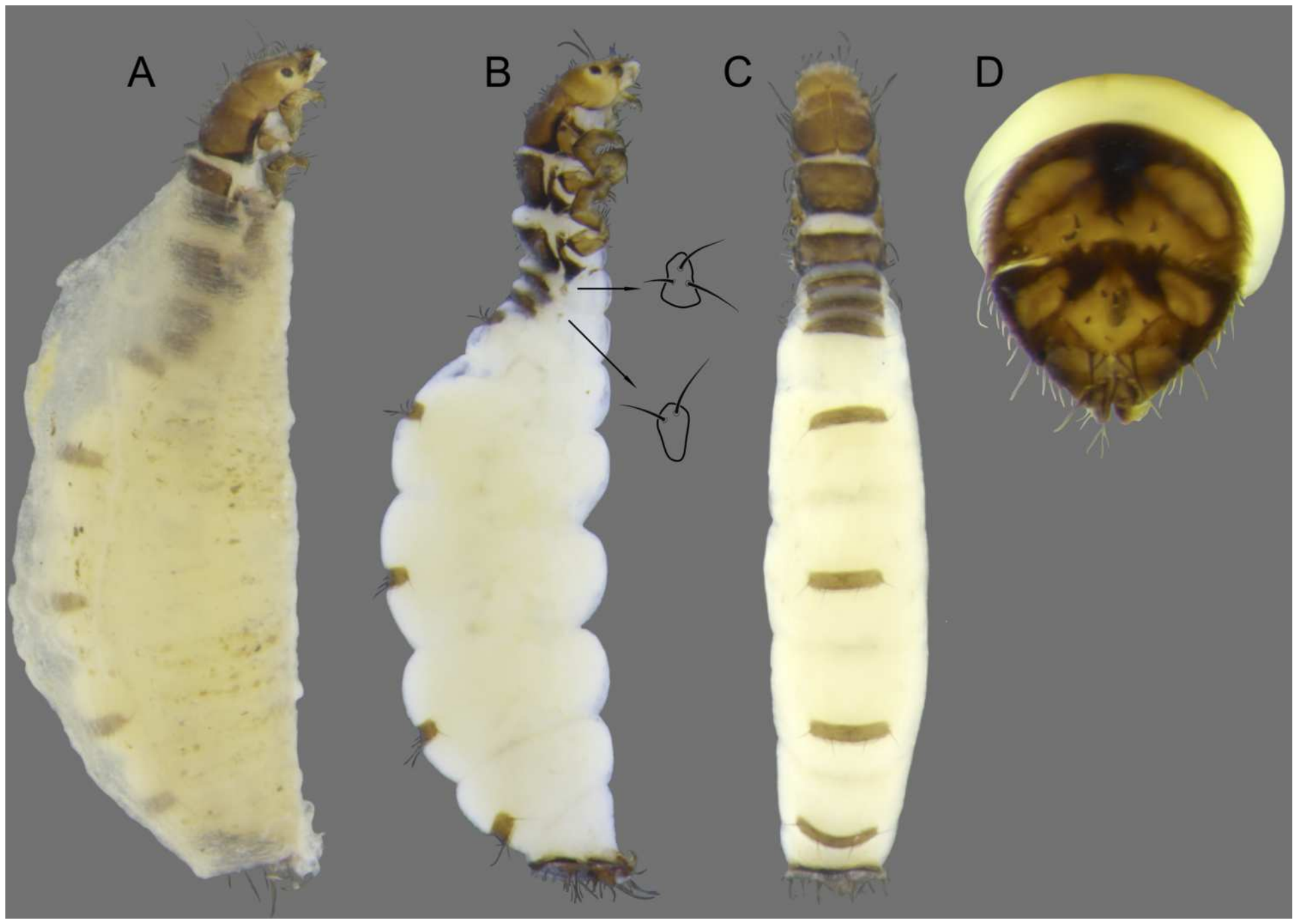

**Figure 10** ***Byrsopteryx mamaocllo* sp. nov., larva.** (A) Lateral habitus with case. (B) Lateral habitus without case. (C) Dorsal habitus. (D) Operculum, dorsocaudal view.

Laval case. Length 1.6–2.8 mm ($n = 29$). Made entirely of silk with bits of mineral material incorporated, translucid; slightly compressed laterally; with poorly closed dorsal seam (Fig. 10A); dorsal margin irregular; anterior and posterior openings circular.

**Type material.** Holotype male. PERU: Cusco, 20 rd km W Quincemil, Pte. Saucipata, Río Araza tributary, 13°20′13″S 70°51′15″W 893 m. 22.viii.2012. APM Santos, DM Takiya leg., pinned (MUSM). Paratypes. Same data as holotype, except 5 males, 7 females, pinned (DZRJ); Same data as holotype, except 5 males, in alcohol (MUSM); PERU: Cusco, 19 rd km W Quincemil, Rio Araza tributary, 13°20′10″S 70°50′57″W 874 m. 23-28.viii.2012. APM Santos, DM Takiya leg., 16 males, 8 females, in alcohol (MUSM); same data, except 5 males, in alcohol (DZRJ); same data, except malaise trap, 4 males, in alcohol (INPA); PERU: Cusco, 3 rd km E Quincemil, 13°13′03″S 70°43′40″W 633 m. 20.viii.2012. APM Santos, DM Takiya leg., 1 female, pinned (DZRJ); PERU: Puno, 6 rd km W Mazuko, Pte. La Cigarra 13°08′27″S 70°23′14″W 353 m. 01.ix.2012. APM Santos, DM Takiya leg., 5 females, pinned (DZRJ); same data, except 1 female, in alcohol (MUSM).

**Additional material examined.** Same data as holotype, 3 larvae, in alcohol (MUSM); PERU: Cusco, 19 rd km W Quincemil, Rio Araza tributary, 13°20′10″S 70°50′57″W 874 m. 23–28.viii.2012. APM Santos, DM Takiya leg., 11 larvae, in alcohol (INPA); same data, except 18 larvae, 2 male pharate adult, in alcohol (DZRJ); same data, except 10 larvae, in alcohol (MUSM).

**Etymology.** The species name is an allusion to the Inca goddess Mama Ocllo (used as a noun in apposition), the "mother fertility". According to Inca mythology, Mama Ocllo

was daughter of Apu Inti and Mama Quilla, and she married her brother Manco Capac. Together, Mama Ocllo and Manco Capac founded Cusco and guided the people, enabling the beginning of the Inca civilization.

**Distribution.** Peru (Cusco, Puno).

**Remarks.** In the field, *B. mamaocllo* **sp. nov.** can be recognized from other two Peruvian species of *Byrsopteryx* by the color pattern with a long longitudinal white band on each forewing (Fig. 6). *Byrsopteryx* species show no sexual dimorphism in color pattern (*Harris & Holzenthal, 1994*), which have allowed male–female associations presented here. *Byrsopteryx mamaocllo* **sp. nov.** can be easily distinguished from other *Byrsopteryx* species also by its male genitalia; with segment VIII bearing a pair of short, posterior, spinelike processes on sternum, which are curved inwards and bordered by rounded, membranous structure (Figs. 8A, 8B); and a pair of short, heavily sclerotized, dorsolateral processes (Figs. 8A, 8D). See further comments on above Remarks of *B. inti* **sp. nov.** Females of *B. mamaocllo* **sp. nov.** also differ from those of other *Byrsopteryx* species by genitalia with the vaginal apparatus only slightly sclerotized, with anterior region rounded, and a posterior region mostly membranous (Fig. 9B).

Larvae of *B. mamaocllo* **sp. nov.** and of *B. mancocapac* **sp. nov.** are very similar to those previously described for other *Byrsopteryx* species, including general aspect of the case, the setation pattern of the head, and the sclerites. General aspects of *Byrsopteryx* larval morphology have been described in detail by *Holzenthal & Harris (1991)* and *Santos & Nessimian (2010)*. Considering the two Peruvian larvae described here, *B. mamaocllo* **sp. nov.** differ from *B. mancocapac* **sp. nov.** by the: (1) almost circular operculum with two pairs of oval lighter areas, a pair on tergite VIII and another on IX (Fig. 10D), in *B. mancocapac* **sp. nov.** larvae the operculum is more rectangular; (2) case generally smaller than larva (Fig. 10A), in *B. mancocapac* **sp. nov.** its case is as long as larva; and (3) abdominal segment II with small pleurite bearing two setae (Fig. 10B), whereas in *B. mancocapac* **sp. nov.**, it has three setae.

## *Byrsopteryx mancocapac* new species

urn:lsid:zoobank.org:act:1D7BEBF4-38D5-4A54-8545-A48B7A066E4A

- *Byrsopteryx* sp. PE3 *Santos, Nessimian & Takiya, 2016*:461. Phylogenetic placement.

Figs. 11–14

**Description.** Adult male. *Coloration*. General color dark brown; head dorsum and mesoscutum densely covered with black setae, with no white setae; forewing with four distinct maculae of white setae: a longitudinal oval band at base of medial area, a trapezoidal macula near midcostal margin, a subapical spot near posterior margin of wing, and an apical spot, spreading towards costal margin (Fig. 11). *Length*. Total length 2.6–3.6 mm ($n = 23$). *Head*. Unmodified. Antennae 18-articulated. Ocelli 3. Maxillary palpi 5-articulated; articles I and II very short and globular, article III very long, as long as articles IV and V combined; articles IV and V with similar lengths, each one about twice as long as wide. *Thorax*. Forewing venation strongly reduced; with distinct line of weakness; basal lobe apparently absent; semi-dome or other wing modification absent; retinaculum distinct (Fig. 12A). Hind wing venation strongly reduced; frenulum distinct, with two rows of three to five

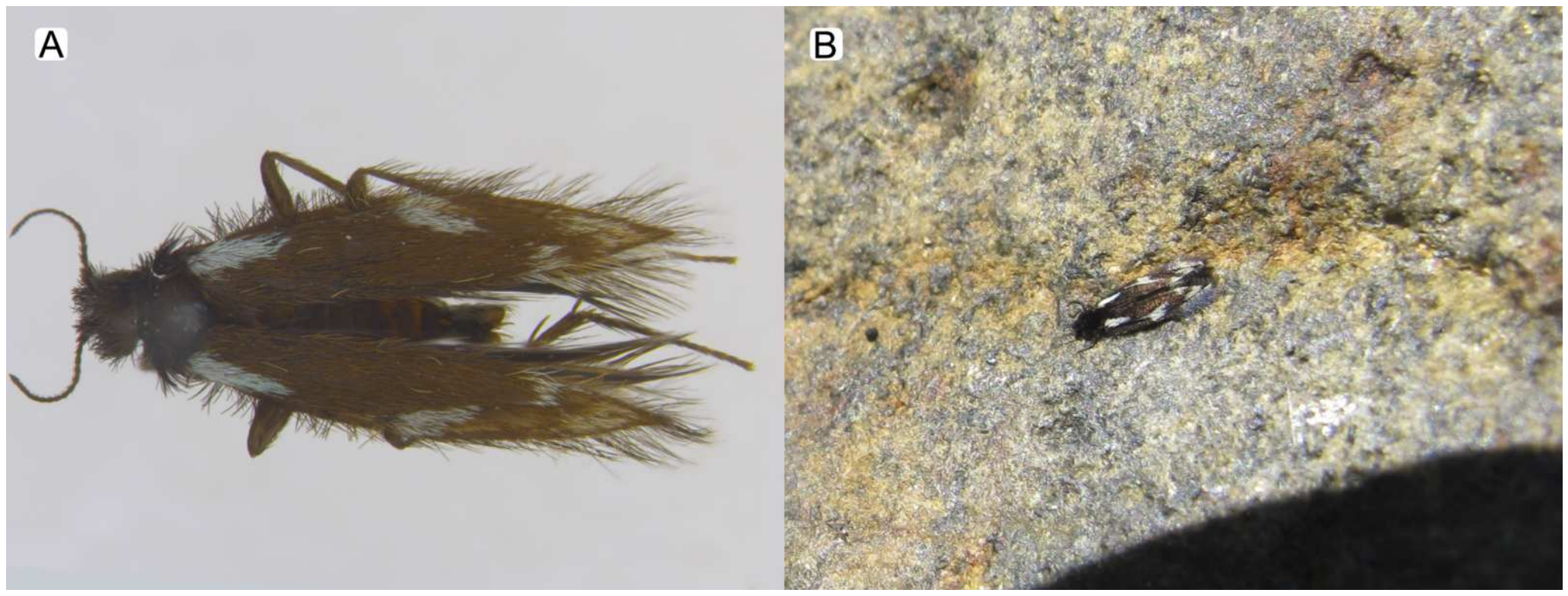

**Figure 11** ***Byrsopteryx mancocapac* sp. nov., adult.** (A) Paratype male (pinned), dorsal habitus. (B) Live adult on a rocky surface.

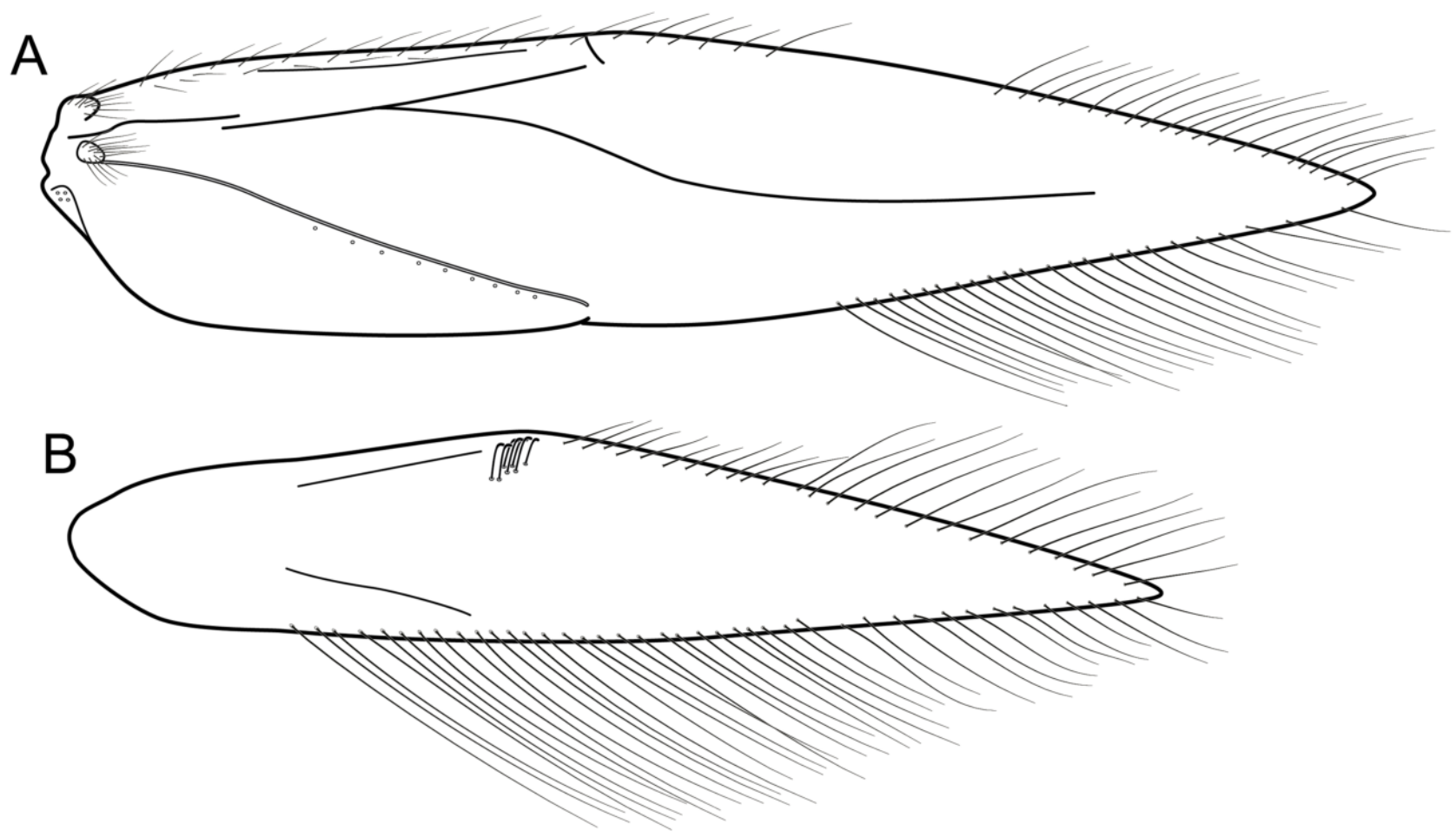

**Figure 12** ***Byrsopteryx mancocapac* sp. nov., male wings.** *Byrsopteryx mancocapac* **sp. nov.**, male wings. (A) Forewing. (B) Hind wing.

short, hooked setae (Fig. 12B). Tibial spur formula 0, 3, 4. *Abdomen*. Segment VII without ventromesal process. *Male genitalia*. Segment VIII shorter dorsally than ventrally. Sternum with deep mesal incision posteriorly, forming pair of platelike lobes in ventral view, each one with rounded corners and bearing longer setae on the posterolateral quadrant and smaller setae surrounding the internal margin (Fig. 13A). Tergum with transverse row of long and stout setae near posterior margin (Fig. 13B). Segment IX recessed within segment VIII, projecting anteriorly through posterior portion of segment VII; with deep mesal incision anteriorly in dorsal and ventral views (Figs. 13A, 13B); narrowing anteriorly in lateral view (Fig. 13C); open dorsally; with pair of small, rounded posterior projections in ventral view, each one with acute beak on internal margin (Fig. 13A). Subgenital plate with lateral arms converging posteriorly to a single darkened apex, strongly downturned in lateral view (Fig. 13C); with strong horn-like process emerging from base of each arm

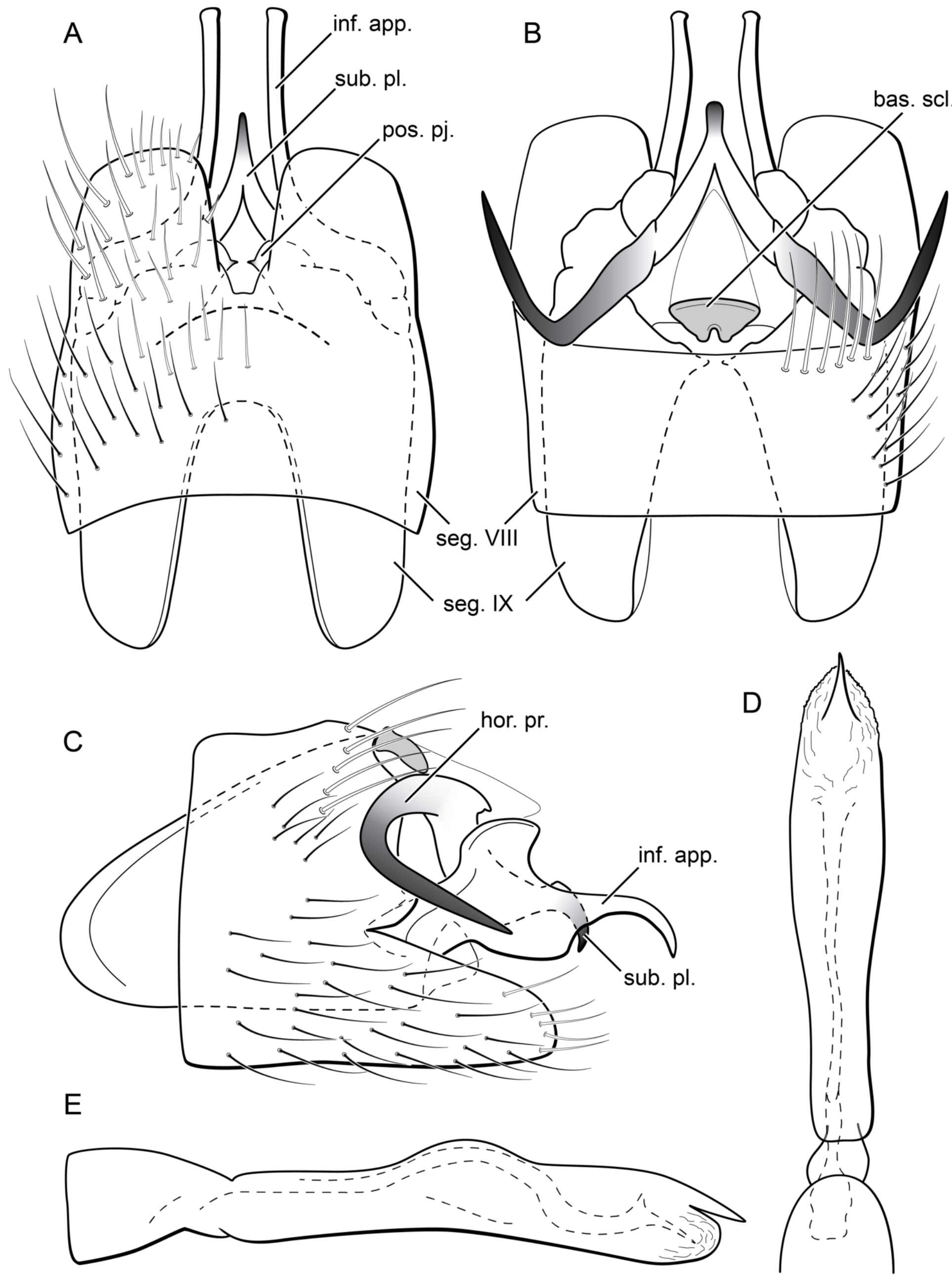

**Figure 13** ***Byrsopteryx mancocapac*** **sp. nov., male genitalia (holotype).** (A) Ventral view. (B) Dorsal view. (C) Lateral view. (D) Phallus, dorsal view. (E) Phallus, lateral view. Abbreviations: seg., segment; pos. pj., posterior projection (segment IX); ter. X, tergum X; bas. scl., basal sclerite (tergum X); sub. pl., subgenital plate; hor. pr., horn-like process (subgenital plate); inf. app., inferior appendage.

(Fig. 13C). Inferior appendages positioned dorsolaterally; elongate, slender, almost parallel in ventral and dorsal views (Figs. 13A, 13B); in lateral view, each one with dorsal platelike projection at basal portion, narrowing to apex and downturned (Fig. 13C). Tergum X with a basal trapezoid sclerite, with shallow incision in anterior margin in dorsal view (Fig. 13B);

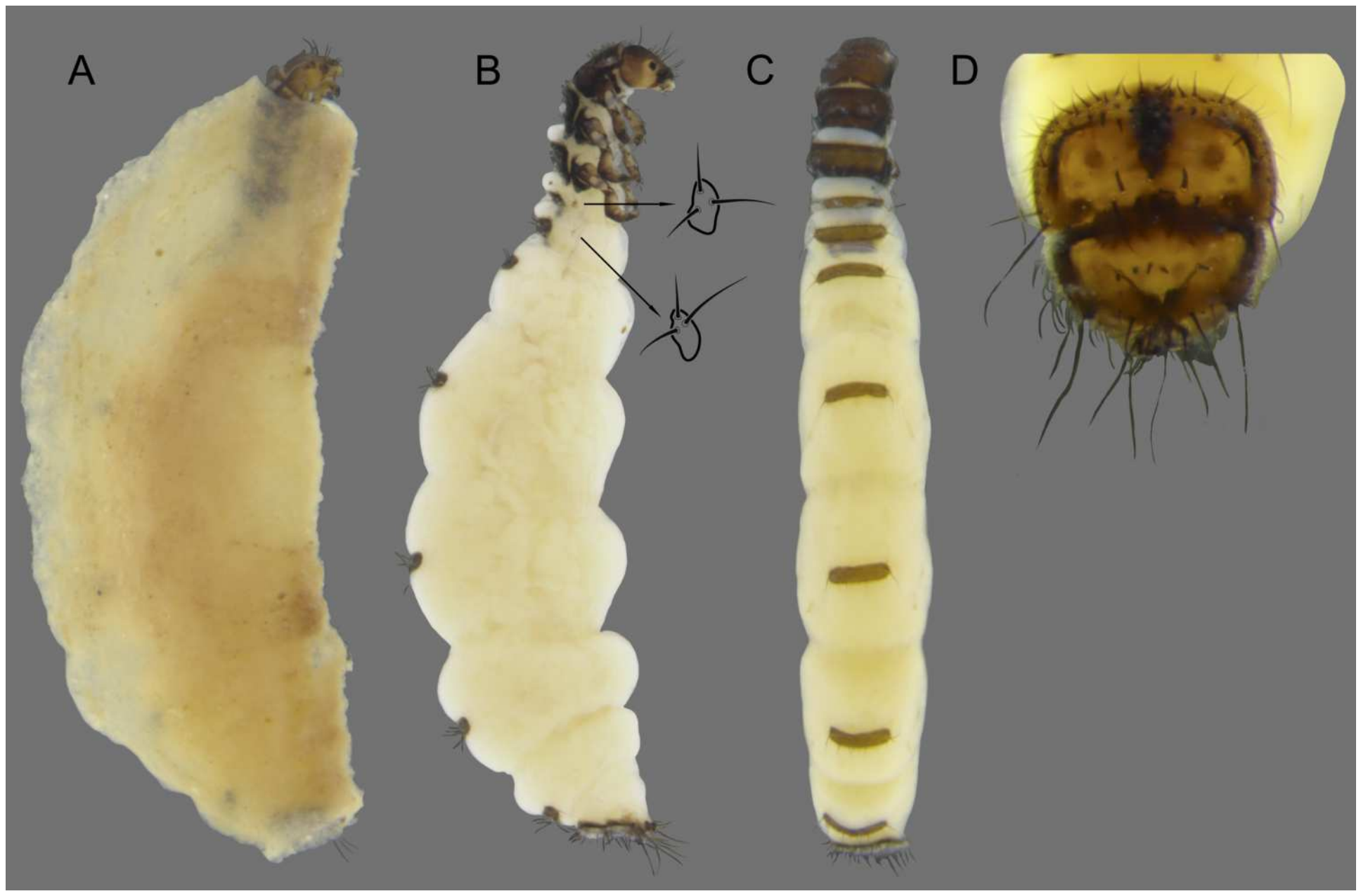

**Figure 14** ***Byrsopteryx mancocapac*** **sp. nov., larva.** (A) Lateral habitus with case. (B) Lateral habitus without case. (C) Dorsal habitus. (D) Operculum, dorsocaudal view.

posterior portion conical and membranous. Phallus tubular, with slight constriction between basal and apical portions; basal portion short, less than half length of apical portion; with an acute sclerotized apical projection (Figs. 13C, 13D); membranous at apex (Fig. 13C).

Adult female. Unknown.

Larva (final instar). Length 2.0–3.8 mm ($n = 28$). Head brown to dark brown, unpigmented around eyes; quadrangular; frontoclypeal and coronal sutures indistinct; dorsum with reticulate pattern formed by microscopic setae; setae 9 very long; antennae 1-articulated, apical seta indistinct; mandibles without distinct teeth. Thoracic nota strongly sclerotized, brown, with lateral and posterior margins dark brown, with short stout setae; pro- and mesonota with transverse mesal sclerotized ridge, less developed on metanotum; pronotum with middorsal ecdysial line, pair of anterolateral depressed areas (Fig. 14C); meso- and metanota slightly broader than long, without middorsal ecdysial lines (Fig. 14C); meso- and metathoracic pleural sclerite large, sclerotized (Fig. 14B); ventrally, prothorax with two pairs of thin, elongate sclerites, which converge medially; with ventral thin intersegmental sclerite between meso- and metathorax; thoracic legs short, stout, similar in size, shape, and setation (Fig. 14B). Abdomen slightly compressed (Fig. 14C); all abdominal segments with sclerotized tergites: segments I and II with large tergite, covering most of dorsal area of the segment (Fig. 14C); segments III-VII with transverse tergite, lightly sclerotized, bearing many setae; segments VIII and IX with platelike tergite, heavily sclerotized, setose, forming a dorsal, subrectangular operculum (Fig. 14D), which has a typical color pattern, with tergites brown, with mesal, longitudinal darker area on tergite

VIII and darker line near anterolateral margins, and darker border on anterior and lateral margins of tergite IX (Fig. 14D); operculum closing posterior opening of case. Pleurites of segments I and II small, each one with three setae (Fig. 14B, in detail); those of segments III-VIII absent, but region with two small setae each one. Anal proleg short, setose with pair of long posterior setae; anal claw stout, curved to approximately 90° , with basal peglike setae.

Laval case. Length 1.8–4.0 mm ($n = 28$). Made entirely of silk with bits of mineral material incorporated, slightly compressed laterally, with poorly closed dorsal seam (Fig. 14A); dorsal margin irregular; anterior and posterior openings circular.

**Type material.** Holotype male. PERU: Cusco, 20 rd km W Quincemil, Pte. Saucipata, Río Araza tributary, 13°20′13″S 70°51′15″W 893 m. 22.viii.2012. APM Santos, DM Takiya leg., pinned (MUSM). Paratypes. Same data as holotype, except 9 males, pinned (DZRJ); same data as holotype, except 7 males, in alcohol (MUSM); same data as holotype, except 1 male, in alcohol (DZRJ). PERU: Cusco, 3 rd km E Quincemil, 13°13′03″S 70°43′40″W 633 m. 20.viii.2012. APM Santos, DM Takiya leg., 3 males, in alcohol (DZRJ). PERU: Puno, 6 rd km W Mazuko, Pte. La Cigarra, 13°08′27″S 70°23′14″W 353 m. 01.ix.2012. APM Santos, DM Takiya leg., 4 males, pinned (DZRJ), 4 males, pinned (MUSM); same data, except, 3 males, in alcohol (INPA), 6 males, in alcohol (DZRJ).

**Additional material examined.** Same data as holotype, except 4 larvae, in alcohol (MUSM), 3 larvae, in alcohol (INPA). PERU: Cusco, 19 rd km W Quincemil, Rio Araza tributary, 13°20′10″S 70°50′57″W 874 m. 26.viii.2012. APM Santos, DM Takiya leg., 8 larvae, in alcohol (DZRJ), 7 larvae, in alcohol (MUSM), 2 larvae, in alcohol (INPA). PERU: Puno, 6 rd km W Mazuko, Pte. La Cigarra, 13°08′27″S 70°23′14″W 353 m. 01.ix.2012. APM Santos, DM Takiya leg., 4 larvae, in alcohol (DZRJ); same data, except 15 larvae, in alcohol (DZRJ); 15 larvae, in alcohol (INPA); 18 larvae, in alcohol (MUSM).

**Etymology.** The species name is an allusion to Manco Capac (used as a noun in apposition), son of Inti and Mama Quilla, according to some legends. Manco Capac and Mama Ocllo, his sister and wife, were sent to Earth to find the best place to begin a civilization. They travelled to the region which later became Cusco, where they established the center of the Inca Empire.

**Remarks.** *Byrsopteryx mancocapac* **sp. nov.** is very distinctive from the previous and other species described in the genus. In the field, this new species can be easily recognized from others occurring at same site by its coloration, whereas adults from both *B. inti* **sp. nov.** and *B. mamaocllo* **sp. nov.** have head and mesoscutum densely covered by white setae (Figs. 3, 6), and adults of *B. mancocapac* **sp. nov.** show only black setae on head and mesoscutum (Fig. 11). Furthermore, males of *B. mancocapac* **sp. nov.** do not show the wing modification which is seen in males of both *B. inti* **sp. nov.** and *B. mamaocllo* **sp. nov.** Male genitalia are also unique among *Byrsopteryx* species, particularly due to the conspicuous spine-like, curved processes projecting from dorsal region of subgenital plate (Figs. 13B, 13C). These processes superficially resemble those of *B. loja* *Harris & Holzenthal, 1994*, but in *B. loja* the paired processes arise from segment IX, instead of subgenital plate, as in the new species described here. In addition to the spine-like processes, this new species is also easily distinguished from *B. loja* and other *Byrsopteryx* species by sternum VIII with posterior

margin divided into two plate-like lobes (Fig. 13A). Although a relatively high number of males and larvae of *B. mancocapac* **sp. nov.** were collected, females were not found.

Larvae of this new species show typical features of *Byrsopteryx*, for example, lateral depressed areas on pronotum, tergites VIII and IX large, forming an operculum, and case made of silk, poorly sealed dorsally (Fig. 14). As mentioned before, general aspects of larval morphology of this new species are similar to those previously described for other *Byrsopteryx* species (*Holzenthal & Harris, 1991*; *Santos & Nessimian, 2010*). Comparing with larvae of *B. mamaocllo* **sp. nov.**, the only other *Byrsopteryx* larva known from Peru, the larvae of *B. mancocapac* **sp. nov.** can be recognized by the following features, as mentioned before: (1) operculum subrectangular, with rounded corners (Fig. 14D); (2) case usually as long as larva and, when compared with that of *B. mamaocllo* **sp. nov.**, opaquer (Fig. 14A); and (3) abdominal segment II with a small pleurite bearing three setae, instead of two in *B. mamaocllo* **sp. nov.** (Fig. 14B).

## DISCUSSION

Hydroptilids represent the most diverse caddisfly family (*Morse et al., 2019*), yet many undescribed species remain unnamed in entomological collections. Here, we provided descriptions for three new species of *Byrsopteryx* from Peru. Despite relevant works on Peruvian caddisfly fauna, such as *Flint Jr (1975)*, *Flint Jr (1980)*, *Flint Jr (1996)*, *Flint Jr & Reyes (1991)*, *Flint Jr & Bueno-Soria (1998)*, *Flint Jr & Bueno-Soria (1999)*, *Harris & Davenport (1992)* and *Harris & Davenport (1999)*, no *Byrsopteryx* species had been described or recorded from the country so far. As indicated by studies in other South American countries, we have a very limited knowledge on caddisfly diversity in the Neotropics (*e.g.*, *Ríos-Touma et al., 2017*—Ecuador; *Santos et al., 2020*—Brazil).

*Byrsopteryx* is a relatively small genus, now with 19 described species. The genus has been recorded from Mexico, Central America, north of South America, and south portion of Brazil (*Holzenthal & Calor, 2017*; *Vásquez-Ramos, Osorio-Ramírez & Caro-Caro, 2020*). Species usually show a limited geographic distribution, being associated to waterfalls and/or rocky streams. Adults are commonly seen active during daylight and usually they are not collected in high numbers using light (*Harris & Holzenthal, 1994*) or Malaise traps. During collecting days, we saw many adults and larvae in the localities where we found *Byrsopteryx*, but very few specimens were collected by several days of Malaise trapping in the same site. Interestingly, the three species described here co-occur in these localities, but adults of *B. mamaocllo* **sp. nov.** were much more seen and collected than the other two species, and *B. inti* **sp. nov.** being rarely represented among sampled specimens. In addition, despite the relatively high numbers of specimens of *Byrsopteryx mancocapac* **sp. nov.**, we were not able to associate any female to this species.

Adult males of the species described here are very distinctive morphologically from other *Byrsopteryx* species, but they have typical features of this genus, such as the color pattern, the diurnal behavior, the reduced wing venation, and forewing with a distinct line of weakness, separating the posterobasal area (*Flint Jr, 1981*). Larvae associated for two of the new species described here are also very similar to those previously described

in the genus. The three new species, though unnamed, were previously included in the phylogenetic analyses presented by *Santos, Nessimian & Takiya (2016)*, which recovered the genus as monophyletic, as indicated earlier by *Harris & Holzenthal (1994)*.

In the present phylogenetic hypothesis *B. mamaocllo* **sp. nov.** grouped with *B. inti* **sp. nov.**, although in the Bayesian analysis of Leucotrichiinae based on combined data (*Santos, Nessimian & Takiya, 2016*), *B. mamaocllo* **sp. nov.** was recovered as sister to *B. mirifica*, the type species. However, the latter clade was not recovered by parsimony analysis of the same dataset nor it was statistically supported, possibly due to the lack of molecular data for *B. mirifica*. Furthermore, in *Santos, Nessimian & Takiya (2016)*, the clade *B. mamaocllo* **sp. nov.** + *B. mirifica* was found related to *B. inti* **sp. nov.** in both analyses and with moderate support. The sister group relationship of these two new species is also herein supported by strong morphological evidence (not shared by *B. mirifica* according to *Harris & Holzenthal, 1994*): the presence of a semi-dome thickening on male forewings (Figs. 5A, 6A), a feature unique among *Byrsopteryx* species. Females of *Byrsopteryx inti* **sp. nov.** are unknown, but those of *Byrsopteryx mamaocllo* **sp. nov.** do not show this wing modification, and since this structure is associated with very long and specialized setae (Figs. 4A, 7A), it could be related to pheromone communication. Species of Leucotrichiinae can show different wing modifications, but they are commonly seen among species in the tribe Leucotrichiini. Among the Alisotrichiini, wing modifications are found in species of *Cerasmatrichia Flint Jr, Harris & Botosaneanu (1994)*, with more sclerotized or inflated areas, but the feature described here for these two new species is unique.

So far, only four *Byrsopteryx* species had their larvae described (*Holzenthal & Harris, 1991*; *Santos & Nessimian, 2010*) and here we describe larvae for two of the new species. These larvae show typical features known for the genus, particularly the case poorly sealed dorsally; prothorax with a pair of anterolateral depressed areas; and all abdominal segments with distinct tergites, with tergites VIII and IX forming an operculum to close posterior opening of the case. Both larvae herein described were associated to adult males based on comparison of COI sequences, and additionally, pharate adults were assigned to *B. mamaocllo* **sp. nov.** due to the general aspect of the case. Caddisfly larvae are conspicuous components of freshwater ecosystems and are much more used in ecological or biomonitoring studies than adults. However, *Pes et al. (2018)* pointed out that only 9.0% of Neotropical caddisfly species have their immatures described, an indicative that more taxonomic studies should be developed in this diverse and poorly known region, in particular, those including larva-adult associations.

## CONCLUSIONS

The present paper is a contribution to the knowledge of Neotropical caddisflies, providing descriptions of three new species of *Byrsopteryx* from Peru, the first formal descriptions of this genus for the country. We were able to associate larvae and adults of two new species based on comparison of a fragment of COI gene. Combining different sources of information, *e.g.*, morphological and molecular data, results in a better understanding of biodiversity, especially of a megadiverse region. The present study highlights the demand

for taxonomic work on caddisflies from the Neotropics, a region increasingly threatened by deforestation due to urbanization and uncontrolled exploration of natural resources.

## ACKNOWLEDGEMENTS

We thank Dr. Isabela C. Rocha and two anonymous referees for comments that improved this manuscript. Peruvian collecting permit was obtained with the help of G. Melo and A. Asenjo (UFPR) and we are very grateful for field assistance provided by J. Rafael (INPA) and R. Cavichioli (UFPR).

### Funding

This work was supported by Fundação Carlos Chagas Filho de Amparo à Pesquisa do Estado do Rio de Janeiro (FAPERJ, Proc. E-26/010.002252/2019). Daniela Maeda Takiya was supported through a research productivity from Conselho Nacional de Desenvolvimento Científico e Tecnológico (CNPq, Proc. 313677/2017-4) and a Cientista do Nosso Estado from FAPERJ (Proc. E-26/202.672/2019) fellowships. The funders had no role in study design, data collection and analysis, decision to publish, or preparation of the manuscript.

### Grant Disclosures

The following grant information was disclosed by the authors:
Fundação Carlos Chagas Filho de Amparo à Pesquisa do Estado do Rio de Janeiro: FAPERJ, Proc. E-26/010.002252/2019.
Conselho Nacional de Desenvolvimento Científico e Tecnológico: CNPq, Proc. 313677/2017-4.
Cientista do Nosso Estado from FAPERJ: Proc. E-26/202.672/2019.

### Competing Interests

The authors declare there are no competing interests.

### Author Contributions

- Allan P.M. Santos and Daniela Maeda Takiya conceived and designed the experiments, performed the experiments, analyzed the data, prepared figures and/or tables, authored or reviewed drafts of the paper, and approved the final draft.

### Field Study Permissions

The following information was supplied relating to field study approvals (i.e., approving body and any reference numbers):

Collecting permit number RD-0297-2012-AG-DGFFS-DGEFFS was issued by Dirección General Forestal y de Fauna Silvestre, Republica del Peru.

### DNA Deposition

The following information was supplied regarding the deposition of DNA sequences:

The COI sequences are available in the Supplementary Files and at GenBank: KU094932, KU094939, KU094942, KU094953, KU094974, KU094975, KU094976, KX107513, AF436490, HQ971757, HQ971758, and OK340604 to OK340612.

### Data Availability

The COI sequences are available in the Supplementary File and at Genbank.

### New Species Registration

The following information was supplied regarding the registration of a newly described species:

Publication LSID: urn:lsid:zoobank.org:pub:71531CB9-F919-4DB9-8885-B8207F0A82F4

Byrsopteryx inti sp. nov.: urn:lsid:zoobank.org:act:9B5FAC2D-A98F-4B88-97BC-CE7CA5105535

Byrsopteryx mamaocllo sp. nov.:
urn:lsid:zoobank.org:act:2F8A72D9-FE9C-4C82-BFA4-26238FBEEBB7

Byrsopteryx mancocapac sp. nov.:
urn:lsid:zoobank.org:act:1D7BEBF4-38D5-4A54-8545-A48B7A066E4A

### Supplemental Information

Supplemental information for this article can be found online at http://dx.doi.org/10.7717/peerj.12645#supplemental-information.

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
