# Peer review of "Three new species of Byrsopteryx Flint microcaddisflies from Peru (Insecta: Trichoptera) including DNA-based larval associations"

_PeerJ, doi:10.7717/peerj.12645_

## Round 0.1 · original submission · Major Revisions

Dear Drs. Santos and Takiya:

Thanks for submitting your manuscript to PeerJ. I have now received three independent reviews of your work, and as you will see, the reviewers raised some minor concerns about the research. Despite this, these reviewers are optimistic about your work and the potential impact it will lend to research on Trichoptera systematics. Thus, I encourage you to revise your manuscript, accordingly, taking into account all of the concerns raised by the reviewers.

While the concerns of the reviewers are relatively minor, this is a major revision to ensure that the original reviewers have a chance to evaluate your responses to their concerns.

Please note that Reviewers 2 &B 3 kindly provided a marked-up version of your manuscript.

Please consider adding the illustrations desired by the reviewers.

I look forward to seeing your revision, and thanks again for submitting your work to PeerJ.

Good luck with your revision,

-joe

Reviewer 1 ·

Basic reporting

English is clear and unambiguous, easily understood throughout the paper.

Sufficient background and context is provided, literature references are appropriate.

Article structure is appropriate for this field, all figures and tables appropriately included. I would actually highly recommend one additional figure: a photo of the forewing showing the semi-dome modification described by the authors, in addition to the already included illustration. This morphological feature has not been observed in any other species in this genus, nor, to my knowledge, other genera within the family. A photo of the feature would be immensely helpful for other researchers and, based on the photos already included of genitalia and larvae, the authors should be capable of producing a photo of a cleared wing mount.

Results and conclusions are appropriate.

Experimental design

Methods well described, all aspects thoroughly detailed.

Research fills an identified knowledge gap, describing additional biodiversity for a hyper-diverse yet under-studied faunal group in a biogeographic region in need of further study. Observations are very complete, based on both morphological and molecular data, and including male, female, and larval diagnoses.

Validity of the findings

All underlying data have been provided and are being stored in the appropriate repositories.

Conclusions are appropriate in scale.

Additional comments

A very thorough description of three new species, a well-structured and well-written paper. My only recommendation would be the additional figure that I mentioned above, under 'Basic Reporting'.

Reviewer 2 ·

Basic reporting

Ref.:  Ms. No. 66094v1
Title: Three new species of Byrsopteryx Flint microcaddisflies from Peru (Insecta: Trichoptera) including DNA-based larval associations

The manuscript constitutes an important contribution to the caddisfly taxonomy, especially on Peruvian fauna. Following the proposed review criteria, I should highlight some points:
1- The text is clear, and the English is completely understandable.
2- The theoretical background is sufficient, and the references are updated.
3- The structure and figures of manuscript are good.
4- The objectives are clear, well defined, and meaningful; the methods are appropriated and can be considered innovative to taxonomy of the order; and the results are very relevant, including the molecular association of adults-immature stages.
5- Corrections and suggestions are in the manuscript file.
The manuscript is well written, provides relevant information about the Peruvian caddisfly fauna and, consequently, makes an important contribution to the Neotropical caddisfly taxonomy. In this way, I strongly support the publication of manuscript in the PeerJ.

With best regards,

Experimental design

.

Validity of the findings

.

Annotated reviews are not available for download in order to protect the identity of reviewers who chose to remain anonymous.

·

Basic reporting

(1) If this is the case, make it clear in the text that the larvae of B. mamaocllo sp. nov. and B. mancocapac sp. nov. do not differ significantly from other described (and illustrated) larvae of the genus. Otherwise, I believe it is necessary to include the illustration of the larvae in the manuscript, as not all the structures described can be seen in the photographs.
(2) To make the manuscript easier to read, I strongly suggest naming and indicating the main structures of the male genitalia in the figure of at least one of the new species.

Experimental design

(1) The method used to associate larvae and male adults was described in detail in the manuscript, but nothing was said about how the association of females was carried out. It is necessary to emphasize that all species described in this work occur in the same localities, therefore, it is not advisable to base the male-female association by co-occurrence.
(2) Describe in the material and methods for which structures of the male genitalia a terminology different from that of Harris & Holzenthal (1994) was adopted.

Validity of the findings

no comment

Additional comments

The work presents a clear and objective writing with small suggestions for changes. Major comments refer to elements that need further clarification to ensure that other readers can easily understand the work. The illustrations are in excellent quality. I congratulate the authors for their efforts to use different sources of information to associate larval and adult stages.

---

## Round 0.2 · accepted · Accept

Dear Drs. Santos and Takiya:

Thanks for revising your manuscript based on the concerns raised by the reviewers. I now believe that your manuscript is suitable for publication. Congratulations! I look forward to seeing this work in print, and I anticipate it being an important resource for groups studying Trichoptera systematics. Thanks again for choosing PeerJ to publish such important work.

Best,

-joe